# Chromatin retained *MUSHER* lncRNA integrates ABA and DOG1 signalling pathways to enhance Arabidopsis seeds dormancy

Sebastian Przemyslaw Sacharowski [1], Michal Krzyszton [1,6], Lien Brzezniak [1,6], Karol Jerzy Rogowski [2], Miguel Montez [3], Katarzyna Rosol[1], Magdalena Wrona [1], Ruslan Yatusevich [1], Veena Halale Manjunath[1,4], Lukasz Szewc[5], Artur Jarmolowski[5] & Szymon Swiezewski [1] ✉

Many plant lncRNAs regulate gene expression by binding to chromatin, but how they are retained at the target *loci* is unclear. We identify a chromatin-localized lncRNA - *MUSHER*, which activates two parallel regulatory pathways to increase Arabidopsis seed dormancy. *MUSHER* is upregulated in response to high temperatures, contributing to the induction of secondary dormancy. It promotes *DOG1* expression by recruitment of the CPSF complex to enhance the proximal cleavage and polyadenylation at the *DOG1* transcript. It also increases ABA sensitivity in seeds by activating *PIR1* gene transcription. These genes, located on different chromosomes, are both bound by *MUSHER*, despite lacking sequence homology. The chromatin association of *MUSHER* enables the integration of the DOG1- and ABA pathways to adjust seed germination timing. Additionally, *MUSHER* and other lncRNAs interact with U1 snRNP, which is required for their chromatin localisation, revealing a role for U1 snRNP in plants.

Multiple long non-coding RNA (lncRNA) are implicated in gene expression regulation in the nucleus by binding to chromatin[1]. Well-known examples of chromatin-bound RNA include *XIST* spreading on the whole X-chromosome and involved in its inactivation in mammals[2,3], and *MALAT1*, tightly bound to chromatin and involved in gene expression and nuclear architecture regulation in humans[4]. There are also many reports of plant lncRNA functioning at the chromatin. This includes *APOLO*, involved in remodelling chromatin structure[5], *SVALKA* proposed to act through induction of RNA polymerase II (Pol II) transcriptional collisions[6] and *PUPPIES*, a lncRNA with a role in delaying Arabidopsis seed germination by controlling Pol II pausing at

*DOG1*[7]. *APOLO* was shown to be retained at *trans*-target *loci* via the formation of R-loops[5]. Probably the best-described plant lncRNA *COOLAIR*, involved in flowering time regulation[8,9], has been shown to form a cloud around its locus[10]. Many lncRNAs functioning in the nucleus are involved in environmental sensing in plants. *COOLAIR* regulates vernalisation[9], SVALKA is implicated in cold acclimation in seedlings[6], *ARTA* enhances ABA response[11], *1GOD* affects drought resistance in mature plants[12], *PUPPIES* regulates seed response to salt[7] and *HID1* promotes germination[13]. While an RNA structural element strongly influences *COOLAIR* chromatin association[14], a broader mechanism for lncRNA retention at chromatin remains elusive.

[1]Laboratory of Seed Molecular Biology, Institute of Biochemistry and Biophysics PAS, Warsaw, Poland. [2]ElementZero Biolabs, Berlin, Germany. [3]Department of Cell and Developmental Biology, John Innes Centre, Norwich Research Park, Norwich, UK. [4]Doctoral School of Molecular Biology and Biological Chemistry, Institute of Biochemistry and Biophysics PAS, Warsaw, Poland. [5]Department of Gene Expression, Institute of Molecular Biology and Biotechnology, Faculty of Biology, Adam Mickiewicz University, Poznan, Poland. [6]These authors contributed equally: Michal Krzyszton, Lien Brzezniak. ✉e-mail: sswiez@ibb.waw.pl

Recently, a U1-based mechanism of human and mouse lncRNA retention at the chromatin was reported[15]. This is based on the physical binding of U1 to a large fraction of lncRNAs[15]. U1 is a highly conserved, small nuclear RNA (snRNA) that, together with conserved proteins including U1C and U1K70, forms the U1 snRNP[16]. U1 snRNP is primarily involved in the first step of splicing by recognising 5' splice sites[17]. However, U1 is also engaged in several other chromatin-related activities: enhancement of Pol II promoters directionality[18]; boosting the speed of RNA polymerase II (Pol II)[19]; and, conserved in plants, suppression of premature cleavage and polyadenylation in introns, known as telescripting[20,21]. Whether U1 snRNP plays a role in chromatin retention in plants is unknown.

Here, we leverage seed biology regulation to explore the functions of lncRNAs. Two key players in seed dormancy are plant hormone abscisic acid (ABA) and a plant-specific protein−DELAY OF GERMINATION 1 (DOG1). Both DOG1 and ABA inhibit seed germination by promoting dormancy, an ability to postpone germination despite favourable external conditions[22]. This allows seeds to ignore short spells of permissive conditions and align germination with seasonal cues. Seed dormancy is initially established when seeds develop on the mother plant[22]. Once this dormancy, known as primary, is released, exposure of imbibed seeds to unfavourable conditions can result in the establishment of secondary seed dormancy[23]. In many plant species, including Arabidopsis, temperature is a key environmental factor required for the efficient establishment of secondary dormancy[24].

DOG1 protein has been suggested to act as part of a thermal-sensing mechanism inducing secondary dormancy[25], but the precise mechanism of how temperature is integrated into the DOG1 gene regulation during secondary dormancy establishment is not clear[22]. DOG1 expression is positively correlated with dormancy strength among Arabidopsis accessions, and it has been identified as a major QTL for seed dormancy and shown to have a significant association in genome-wide studies (GWAS)[26,27]. DOG1 transcription is extensively regulated, including by two independent lncRNA. DOG1 expression is suppressed by an antisense transcript 1GOD, and activated in response to salt by PUPPIES, a lncRNA expressed in the DOG1 promoter[7,28]. In addition, the selection of the DOG1 proximal polyA site results in the production of shDOG1 (short DOG1), the predominant and functional form of the DOG1 transcript, while the selection of the distal polyA site results in a presumably non-functional and much less abundant lgDOG1 (long DOG1) transcript[29–31]. Multiple regulators of DOG1 proximal/distal polyA site selection have been described[30,32], but DOG1 alternative polyadenylation has so far not been linked to environmental responses.

DOG1 expression is, to a large extent, unaffected by ABA levels, and dog1 mutations only marginally affect ABA levels in seeds[33]. Interestingly, seed sensitivity to ABA can be modified by both environmental factors and mutations in genes involved in the ABA signalling pathway, without affecting basal ABA levels[34]. ABA is sensed in plants by ABA receptors PYR/PYLs, which once bound by ABA, bind to and inactivate several PP2C phosphatases, including PP2CA[35]. One way plants can modify ABA sensitivity is by interfering with the ABA signal transduction pathway. For example, PP2CA interacting RING finger protein 1 (PIR1) and PIR2 have been shown to mediate PP2CA degradation to enhance abscisic acid response[36]. Although dog1 mutants do not exhibit altered ABA sensitivity[27], DOG1 protein has been shown to interact with and inhibit the enzymatic activity of PP2CA in vitro[37,38]. In summary, DOG1 and ABA pathways are closely intertwined at the protein level, but genetic and transcriptomic analyses indicate that DOG1 function independently from ABA[22,27,33].

Here, we describe MUSHER, a lncRNA that integrates ABA and DOG1 pathways on the transcriptional level by enhancing DOG1 and PIR1 expression. Through genetic analyses, we demonstrate that MUSHER acts as a positive regulator of dormancy. Our findings imply that MUSHER responds to increased temperature and is essential for high-temperature-induced secondary dormancy in Arabidopsis. At the molecular level, MUSHER promotes DOG1 mRNA proximal polyadenylation by recruiting the CPSF complex. Additionally, through RNA Pol II recruitment, MUSHER facilitates the expression of PIR1 in trans, thereby enforcing ABA sensitivity. Finally, we show that MUSHER, as well as COOLAIR and PUPPIES, interact with chromatin through U1 snRNP, suggesting the existence of a conserved mechanism of lncRNA-chromatin retention across eukaryotes.

## Results
### Chromatin-localised lncRNA, MUSHER, regulates DOG1 and seed dormancy
To identify lncRNA controlling DOG1, we performed RT-qPCR analysis with primers spanning the DOG1 locus in WT and hen2 mutant seeds (Fig. 1a,b). HEN2 is an RNA helicase that interacts with the exosome complex responsible for the nuclear 3' to 5' degradation of various ncRNAs[39], and its mutation provides a good background to identify unstable lncRNA species[40,41]. We found a hen2-dependent signal near the DOG1 gene 3'end, pointing to transcription of a previously uncharacterised transcript targeted by the exosome in seeds (Fig. 1b). We did not detect increased signals in regions corresponding to PUPPIES and 1GOD transcripts[7,28], indicating that these DOG1 locus-originating lncRNAs are not targeted by the exosome in seeds. 5' and 3'RACE-seq analyses (Fig. 1c, Supplementary Data 1) suggested the existence of an intergenic transcriptional unit between the SnRK3.6 and DOG1 genes, lacking protein-coding potential (Supplementary Data 2). Due to its location downstream of DOG1, we named this putative lncRNA MUSHER.

Measuring this ncRNA level throughout seed development (Fig. 1d) revealed that MUSHER transcripts accumulate during seed maturation concomitant to DOG1 induction. To disrupt the MUSHER transcriptional unit, we used two T-DNA insertion lines, msh-1 and msh-2 (Fig. 1a), a Cas9 deletion mutant Δmsh, and two additional lines: msh_dCas9-1 and msh_dCas9-2 with catalytically inactive Cas9 (dCas9). dCas9 acts as a transcriptional repressor by binding to target DNA without inducing cleavage[42]. T-DNA insertion and dCas9 lines, but not the dCas9 controls targeting two intergenomic regions localised 100 kb from DOG1 locus showed reduced MUSHER levels in seeds (Supplementary Fig. 1a,b). Unexpectedly, Δmsh, despite deletion of mapped MUSHER start site, showed ectopic expression of an antisense transcript downstream of DOG1 (Supplementary Fig. 1c and d). Despite this, all tested musher mutants demonstrated reduced DOG1 expression in seeds and increased seed germination (Fig. 1e,f and Supplementary Fig. 1e). The impact of msh-1 and msh_dCas9-1 on DOG1 expression is observed throughout seed maturation (Fig. 1g and Supplementary Fig. 1f). Because of the observed MUSHER-like transcripts in Δmsh we decided to exclude this line from downstream analysis. We generated two MUSHER-complemented lines: msh-1 pMUSHER_1 and msh-1 pMUSHER_2 to determine whether expression of MUSHER under its endogenous promoter in the msh-1 background rescues its low-dormancy phenotype. Both lines exhibited higher DOG1 expression (Supplementary Fig. 1g) and stronger dormancy (Supplementary Fig. 1h,i) than msh-1 mutants, resembling wild-type (WT) plants, further supporting the role of MUSHER in regulating DOG1-dependent dormancy.

Since MUSHER and DOG1 are co-expressed during seed maturation (Fig. 1d), we questioned whether DOG1 expression can have a feedback effect on MUSHER transcription. We observed reduced MUSHER expression in dog1 insertional mutants, dog1-3 and dog1-4, but no effect in a DOG1 large deletion mutant generated using CRISPR/Cas9, dog1-Δ19 (Supplementary Fig. 1a). This suggested that while MUSHER regulates DOG1 expression, MUSHER is not dependent on DOG1 expression. We speculated that the downregulation of MUSHER expression in dog1 insertional mutants could result from local chromatin state changes caused by T-DNA insertion[43]. Furthermore,

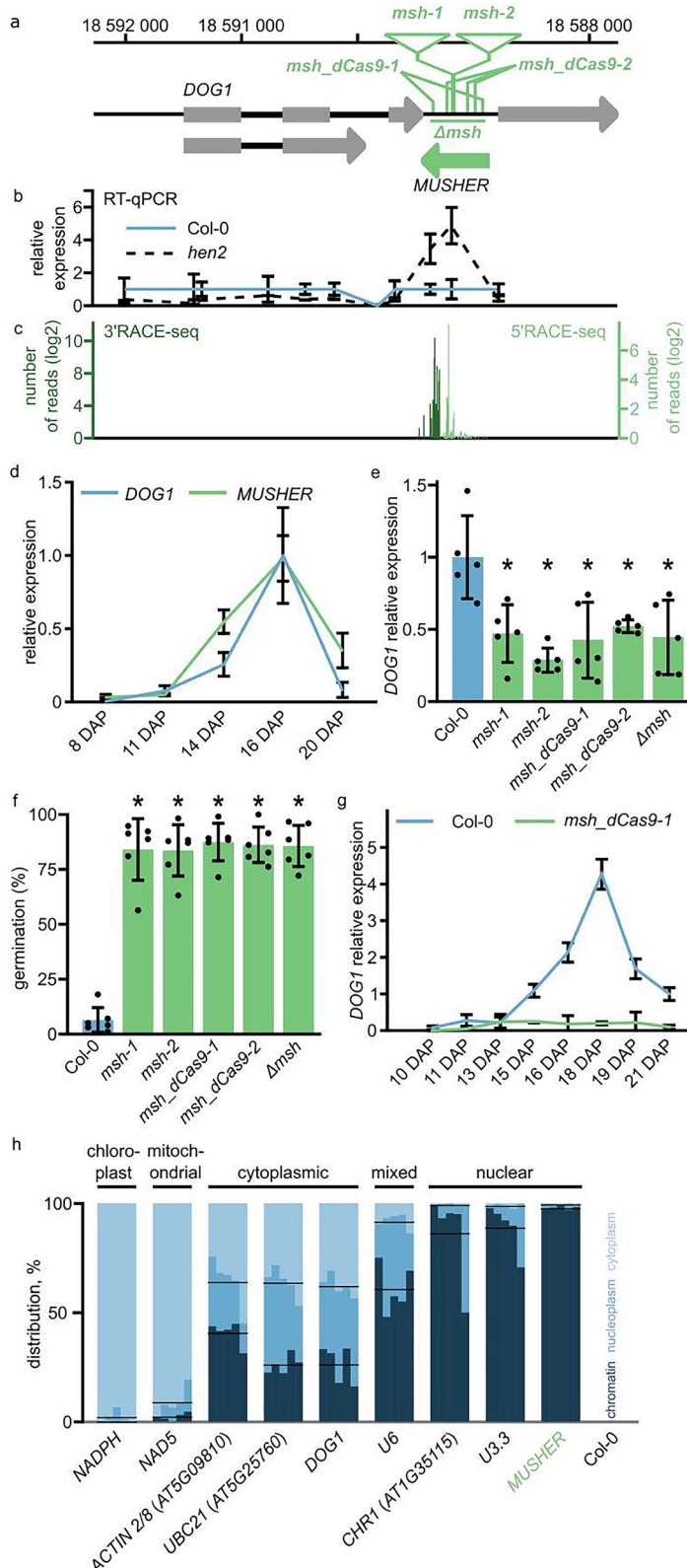

*MUSHER* expression was unaffected by mutations in the downstream gene *SnRK3.6* (Supplementary Fig. 1a), confirming that *MUSHER* is an independent transcriptional unit.

Next, we tested if *MUSHER* accumulates in the cytoplasm or chromatin, which could provide insight into its mode of action. To test *MUSHER* localisation, we used developing seeds, as this is when *MUSHER* and *DOG1* are co-expressed at high levels, and *DOG1* exerts its

function in seed dormancy establishment. We isolated cytoplasmic, nucleoplasmic and chromatin fractions and performed RT-qPCR analysis. Mitochondrial and chloroplast-localised RNA, used as negative controls, are nearly exclusively found in the cytoplasmic fraction. Several tested nuclear-encoded mRNAs showed similar levels in cytoplasm, nucleoplasm and chromatin. U6, a spliceosome component, was detected mainly in chromatin and nucleoplasmic fractions.

**Fig. 1 | *MUSHER* lncRNA acts as a positive regulator of seed dormancy and *DOG1* expression. a** Schematic representation of the *DOG1-MUSHER* genomic region of chromosome 5 in *A. thaliana*. The direction of transcription is indicated with arrowheads. Grey rectangles represent coding exons and introns with thick black lines. The green arrow represents the newly annotated MUSHER transcriptional unit. Numbers along the x-axis on the top indicate genome coordinates (TAIR10). Triangles represent T-DNA insertions of two T-DNA lines (*msh-1* and *msh-2*). The green vertical lines show the positions of multiple guides for *msh_dCas9-1* and *msh_dCas9-2 dCas9* lines. The green horizontal line corresponds to the region deleted using Cas9 (Δ*msh*). **b** Primer walking RT-qPCR analysis across the *DOG1* locus. The position of amplicons corresponds to the *DOG1* locus representation above. Expression levels in the *hen2* mutant were normalised to WT and *UBC21*. Data are presented as mean values of four biological replicates +/− SD. **c** Positions of *MUSHER* transcript ends found in WT seeds using 3' and 5' RACE-seq, coloured in dark and light green, respectively. **d** Expression profiles of *DOG1* and *MUSHER* transcripts during seed maturation in the WT. Expression levels were normalised to *UBC21* mRNA and 16 DAP. The data are shown as the average of four biological replicates for *DOG1* and *MUSHER* 11DAP, five for *MUSHER* 16DAP, and six for *MUSHER* 8,14DAP ± SD. **e** *DOG1* relative expression levels, musher mutants relative to WT in freshly harvested seeds, normalised to *UBC21*. *$p < 0.05$, two-tailed Student's t-test. Data are the mean of five biological replicates, with error bars indicating the standard deviation. **f** Freshly harvested seeds of WT and different *musher* mutants were scored for germination. Germination was defined as radical protrusion and counted 4 d after sowing. Data are the mean of six biological replicates +/− SD ($n = 6$, *$p < 0.05$, two-tailed *t*-test). **g** *DOG1* expression profile during silique development in WT and msh_dCas9-1 plants. Expression levels were normalised to *UBC21* and shown at the indicated number of days after pollination (DAP); Data are presented as mean values of three biological replicates for WT and four for *msh_dCas9-1* +/− SD. **h** Subcellular localisation of selected transcripts in maturing WT Arabidopsis seeds. Each transcript was tested in 5 biological replicates represented as separate bars. Black horizontal lines represent the mean of chromatin and nucleoplasmic fractions contribution of the total amount of indicated transcript.

*MUSHER*, *CHR1* - a transposon-derived RNA and U3.3, involved in co-transcriptional pre-rRNA modification, were predominantly detected in chromatin fraction (Fig. 1h).

In summary, our data showed that *MUSHER* is a lncRNA transcribed downstream of the *DOG1* gene that increases *DOG1* mRNA levels to delay the germination of freshly harvested Arabidopsis seeds. *MUSHER's* chromatin association suggests that it may control *DOG1* expression by binding to its chromatin.

## *MUSHER* lncRNA is required for *DOG1* regulation by CPL1, a Pol II phosphatase

We showed previously that RNA Polymerase II Carboxyl Terminal Domain (CTD) Phosphatase-Like1 (CPL1)-mediated suppression of *DOG1* expression (Supplementary Fig. 2a) requires the 3' end of the *DOG1* locus but not the antisense *1GOD* lncRNA expression from this region[32]. Consequently, the molecular mechanism of CPL1 control over *DOG1* expression remains incompletely understood. Given the effect of *MUSHER* on *DOG1* expression and dormancy, supported by the *MUSHER*-complemented lines (Supplementary Fig. 1g,h), we speculated that CPL1 could act through *MUSHER* regulation. Indeed, multiple *cpl1* alleles showed elevated *MUSHER* transcript level (Supplementary Fig. 2b), suggesting that *cpl1* may act as *MUSHER* lncRNA overexpressor.

To test the requirement of *MUSHER* for CPL1's effect on *DOG1*, we constructed a full-length *DOG1*-luciferase (LUC) fusion with or without *MUSHER* (*pDOG1::LUC-DOG1*[30] and *pDOG1::LUC-DOG1ΔMUSHER*). Transient expression in Arabidopsis seedlings showed that *MUSHER* deletion suppressed *DOG1* upregulation in the *cpl1* plants (Supplementary Fig. 2c).

To cross-validate this, we checked the effect of CPL1 on *MUSHER* locus activity. We generated transgenic plants expressing an internal ribosomal entry site-luciferase gene (IRES-LUC) reporter cassette fused with *MUSHER* and crossed them to *cpl1-7* mutant. The resulting *pMUSHER::MUSHER-IRES-LUC* (*pMUSHER_1*) showed stronger activity in *cpl1-7* when compared to WT, indicating that *MUSHER* is a direct target of CPL1 (Supplementary Fig. 2d). Our data showed that *MUSHER* lncRNA is required for CPL1 to control *DOG1* expression.

Interestingly, inactivation of CPL1 led to preferential expression of proximally over distally polyadenylated *DOG1* mRNA isoform, known respectively as *shDOG1* (*short DOG1*) and *lgDOG1* (*long DOG1*) (Supplementary Fig. 2e). Conversely, *MUSHER* inactivation resulted in reduced *shDOG1* isoform (Supplementary Fig. 2e). Thus, *MUSHER* and CPL1 showed antagonistic effects on the production of *shDOG1*, encoding functional *DOG1* protein and complementing the *dog1* mutant[30]. Importantly, mutating the *cpl1* gene in the *msh* background did not reverse *lgDOG1* polyadenylation preference (Supplementary Fig. 2e), suggesting *DOG1* polyA site selection is directly affected by *MUSHER* and not by CPL1 protein. Taken together, our data showed that CPL1 indirectly affects *DOG1* polyadenylation site selection, by suppressing *MUSHER* expression and activity.

## *MUSHER* promotes *DOG1* proximal polyadenylation by recruiting the CPSF complex

We used 3'RNA-seq (Supplementary Fig. 2f) to confirm that *shDOG1* is decreased in freshly harvested *msh-1* seeds, while *lgDOG1* remains unchanged when compared to WT (Fig. 2a). Calculation of 3'RNA-seq read density assigned to *DOG1* isoforms revealed a significant reduction of *shDOG1/lgDOG1* ratio in *msh-1* (Fig. 2b). Subsequent RT-qPCR analysis confirmed that *shDOG1/lgDOG1* ratio is significantly reduced in all tested *musher* mutants (Fig. 2c).

We previously reported that mutations in pre-mRNA cleavage and polyadenylation factors result in reduced *shDOG1* but not *lgDOG1* expression[30], similar to *MUSHER* knockouts shown here. To confirm changes in *DOG1* mRNA 3'end formation, we performed ChIP-qPCR analysis of the CPSF73, a subunit of the Cleavage Stimulatory Factor complex[44]. We observed a localised decrease in CPSF73 occupancy at the *DOG1* proximal but not distal termination site in *msh-1* and *msh_dCas9-1* mutants (Fig. 2d). Additionally, we performed a genetic analysis of *MUSHER's* interaction with 3'end formation machinery, by crossing *MUSHER* mutants to *esp-1*. ESP encodes an auxiliary component of Cleavage Stimulatory Factor complex (CSTF64L). We used *esp-1* and not *cpsf73* mutants, because the latter do not produce seeds[45]. Analysis of the *msh-1 esp1-2* double mutant showed no additivity in *DOG1* termination site selection defects suggesting that *MUSHER* controls *DOG1* polyA site selection through CPSF complex (Fig. 2e). This was further supported by no change of *MUSHER* expression in the *esp1* mutant (Fig. 2f). Our results implied that *MUSHER* increases *DOG1* expression by enhancing CPSF complex recruitment to the proximal termination site. This is in agreement with the findings in *cpl1-7* mutant, where *MUSHER* overexpression was accompanied by enhanced *DOG1* proximal polyA site selection (Supplementary Fig. 2). Transcription termination can be coupled with Pol II occupancy changes[46]. We, therefore, profiled Pol II occupancy along the *DOG1* locus in the *musher* mutant. This revealed a significant decrease of Pol II level at the *DOG1* proximal termination region but not at the *DOG1* transcription start site (TSS) or the distal polyA site (Fig. 2g). This suggested that *MUSHER*-promoted proximal polyA site selection is not accompanied by changes in *DOG1* transcription initiation but rather represents a result of *MUSHER* direct activity at *shDOG1* gene 3'end.

The localised decrease of Pol II and CPSF complexes occupancy at the *DOG1* proximal termination site indicates that *MUSHER* promotes the proximal termination site selection rather than controls transcription rates. To test this hypothesis, we took an orthogonal approach and analysed the intensity of *DOG1* transcription using single-molecule RNA FISH (smFISH) in developing embryos[7]. We measured the brightness of nuclear foci identified as transcription

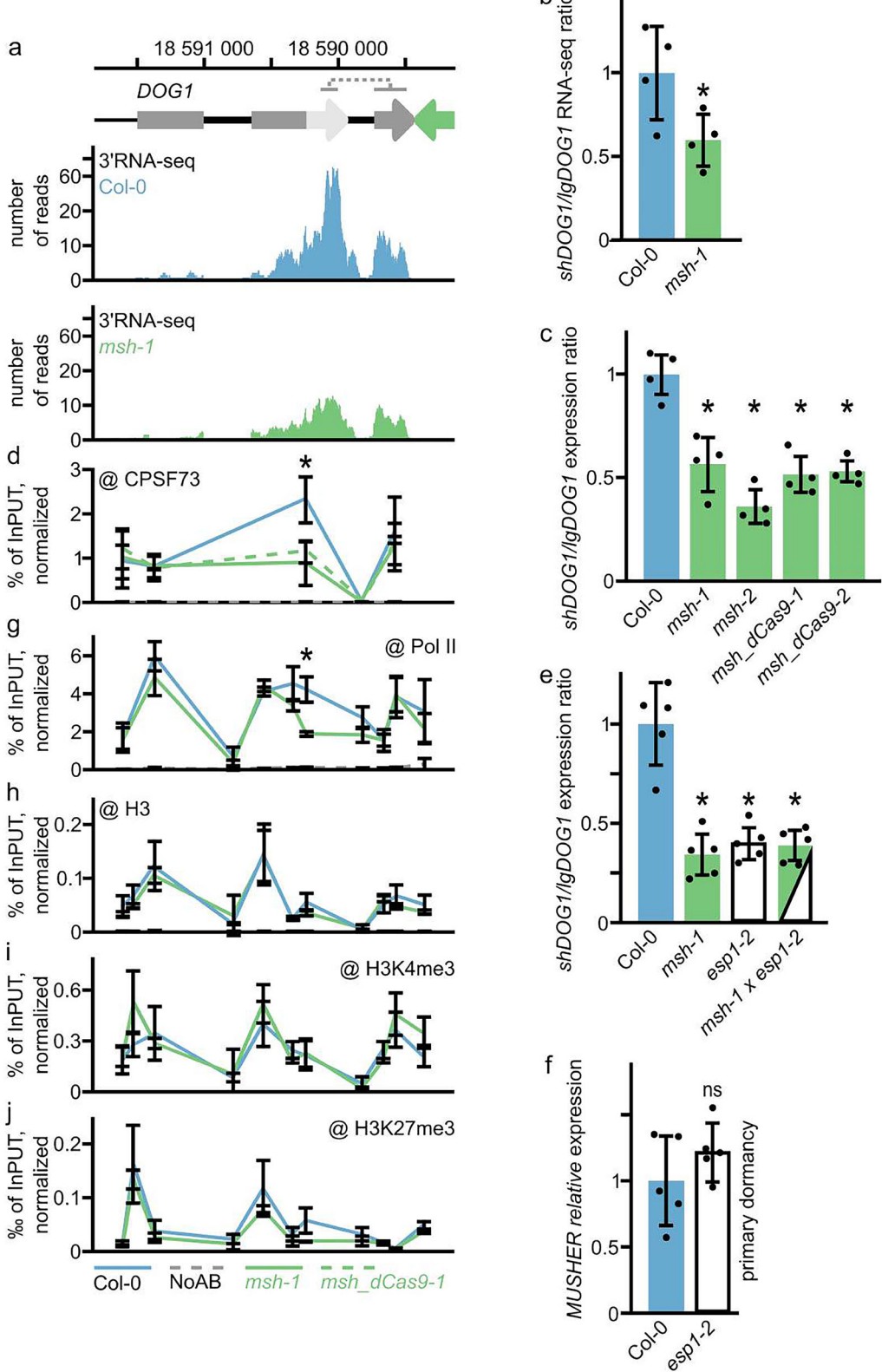

sites, as this is generally considered to correlate with the number of RNA molecules present at transcription sites. We also counted the proportion of cells containing transcription sites, which reflects the frequency of promoter bursting[7]. We collected embryos from WT and *msh-1* siliques at ten time points, corresponding to the experiments showing *DOG1* and *MUSHER* expression during seed maturation (Fig. 1d). *DOG1* smFISH showed a consistent and significant reduction

in the number of both nuclear and cytoplasmic *foci* during seed development in *msh-1* seeds when compared to WT (Supplementary Figs. 3a,b). We observed an increase in *DOG1* transcript number in WT siliques between the 16th and 21st day, which was diminished in *msh-1* (Supplementary Figs. 3a,b). Notably, the *msh* mutant showed no significant change in the intensity of nuclear dots identified as transcription sites (Supplementary Fig. 3c)[7] and only minor changes in the

**Fig. 2 | MUSHER enhances DOG1 proximal polyA site selection. a** Snapshot of 3′ RNA-seq reads for *DOG1* locus in WT and *msh-1* mutant (*n* = 4). Schematic diagram showing *DOG1* gene: exons (grey rectangles), *shDOG1* 3′UTR region (light grey arrow), *lgDOG1* 3′UTR region (grey arrow), and *MUSHER* (green). The regions used to estimate the expression of mRNA isoforms are shown below chromosome coordinates. **b** Ratio of reads corresponding to short and long DOG1 mRNA isoforms calculated for regions marked on the schematic. Data are the mean of four biological replicates; error bars show standard deviations. **c** The ratio of *shDOG1* and *lgDOG1* isoforms in freshly harvested seeds of WT and musher mutants, normalised to *UBC21* and relative to WT. Data are presented as mean values of four biological replicates +/− SD (\**p* < 0.05, two-sided t-test). **d** CPSF73 occupancy was measured by ChIP-qPCR in WT, *msh-1* and *msh_dCas9-1* mutants. Data are presented as mean values of three biological replicates +/− SD (\**p* < 0.05, two-sided *t*-test). **e** The ratio of *shDOG1* and *lgDOG1* isoforms in seeds of WT and *msh-1*, *esp1-2* and double *msh-1 esp1-2* mutants, normalised to *UBC21* and relative to WT (mean of five biological replicates +/− SD, \**p* < 0.05, two-sided *t*-test). The ratio of *shDOG1* and *lgDOG1* isoforms in seeds of WT and *msh-1*, *esp1-2* and double *msh-1 esp1-2* mutants, normalised to *UBC21* and relative to WT (mean of 5 biological replicates, \**p* < 0.05, two-sided *t*-test). **f** *MUSHER* relative expression level in seeds of WT and *esp1-2*, normalised to *UBC21* and relative to WT (mean of five biological replicates +/-SD, ns − not significant, two-sided *t*-test). **g** Pol II occupancy was measured by ChIP-qPCR in WT and *msh-1* mutant. Data are presented as mean values of three biological replicates +/− SD (\**p* < 0.05, two-sided *t*-test). **h** H3, **i** H3K4me3 and **j** H3K27me3 levels were measured by ChIP-qPCR in WT and *msh-1* mutant. H3K4me3 and H3K27me3 values were normalised to H3 and to *ACTIN7* (*AT5G09810*) gene body. For all ChIP-qPCR experiments presented in this figure: *p*-value from the two-tailed *t*-test, \**p* < 0.05, ns - not significant. Data are the mean of four biological replicates, with error bars indicating the standard deviation. H3, H3K4me3 and H3K27me3 (**h–j**) ChIP were done side by side and share NoAB control shown on panel h.

frequency of cells with active *DOG1* transcription (Supplementary Fig. 3d). We therefore conclude that, in agreement with Pol II ChIP (Fig. 2g), smFISH indicated no major changes in *DOG1* transcription initiation in *msh-1* seeds.

Our data suggest that *MUSHER* promotes *DOG1* proximal termination by locally affecting cleavage and polyadenylation complexes and Pol II occupancy at the proximal transcription termination site without affecting *DOG1* transcription levels. All modes of RNA Pol II activity, including transcription termination, are accompanied and modulated by histone posttranslational modifications. Some lncRNAs may recruit specific histone-modelling complexes, including PRC2 involved in H3K27me3 deposition[47,48]. To address the effects of *MUSHER* on the *DOG1* chromatin landscape, we analysed selected active (H3K4me3) or repressive (H3K27me3) chromatin histone marks along the *DOG1* locus. Using ChIP-qPCR, we detected no significant difference in both tested histone modification levels in the *msh-1* mutant throughout the *DOG1* locus (Fig. 2h–j) compared to WT. This result showed that defects at the proximal transcription termination site (TTS) are unrelated to chromatin modifications and agree with no changes in Pol II recruitment for most of the *DOG1* locus. In summary, we propose a model in which *MUSHER* lncRNA specifically enhances *DOG1* proximal termination rather than Pol II recruitment.

## MUSHER enhances secondary dormancy in response to high temperature

*DOG1* is a known regulator of both primary and secondary dormancy[49,50]. Based on the seed's requirement for *MUSHER* to express *DOG1* at high levels and fully develop primary dormancy (Fig. 1), we hypothesised that *MUSHER* may also be involved in secondary dormancy regulation. We used a well-established secondary dormancy induction protocol that consists of prolonged incubation in the darkness at high temperatures followed by germination analysis in permissive conditions[49]. In agreement with defects in primary dormancy, *msh-1* and *msh_dCas9-1* showed weaker secondary seed dormancy, and *cpl1-7*, a mutant with high *MUSHER* expression, showed stronger secondary dormancy when compared to WT (Fig. 3a,b, Supplementary Fig. 4a,b). Consistent with these phenotypes, we observed the down- and up-regulation of the *shDOG1* transcript in *musher* and *cpl1-7* mutants, respectively (Fig. 3c,d).

Elevated temperature is a key environmental factor responsible for secondary dormancy establishment[51]. Consistent with previous studies[7,49,52,53], we showed that *DOG1* expression increases in response to high temperatures during secondary dormancy establishment (Fig. 3e). At 35 °C, *DOG1* expression dropped initially during imbibition but rose to levels observed in dry seeds on the 2nd day of secondary dormancy induction and remained constant for the rest of the experiment (Fig. 3e). Likewise, high temperatures augmented *MUSHER* induction in imbibed seeds (Fig. 3f). *MUSHER* levels were similarly

reduced by imbibition and recovered at 2nd day, but, in contrast to *DOG1*, continued to rise during the entire experiment length without reaching the plateau. This suggests that *MUSHER* is more responsive to secondary dormancy-inducing conditions than *DOG1* (Fig. 3f). Based on this observation, we speculated that *MUSHER* could be responsible for temperature sensing and the incorporation of this signal into *DOG1* regulation.

Next, to analyse the temperature responsiveness of *MUSHER* during secondary dormancy induction, we took advantage of luciferase reporter lines (Supplementary Fig. 2c,d). We induced secondary dormancy at different temperatures (25, 28, and 35 °C) and measured the LUC activity for *DOG1-LUC*, *MUSHER-LUC* and the *DOG1trunc-LUC*, which is devoid of *MUSHER* region. We noted that *MUSHER* and *DOG1* reporters showed similar gradual increases in expression in response to increasing temperatures (Fig. 3g). Interestingly, *DOG1* reporter devoid of *MUSHER* region (*pDOG1::DOG1-LUC truncated*) showed no induction at 35 °C (Fig. 3g), indicating *MUSHER* is required for the high-temperature-driven *DOG1* upregulation. The responsiveness of *MUSHER* to high temperatures, along with *MUSHER* requirement for high temperature-driven *DOG1* upregulation, indicated that *MUSHER* conveys temperature signals during the induction of secondary dormancy. In agreement with these findings, we show the endogenous *shDOG1* expression in wild-type seeds during secondary dormancy establishment is highly induced at 35 °C compared to 28 °C (Fig. 3h), with *lgDOG1* transcript being only slightly affected. This is consistent with the concept of *MUSHER* acting through the polyA-site selection (Fig. 2).

In summary, our data are consistent with a model in which *MUSHER* lncRNA is induced during primary and secondary dormancy and activates DOG1 expression by promoting its proximal polyA site selection both during primary and secondary dormancy. In addition, we showed that during secondary dormancy induction, *MUSHER* lncRNA is required for *DOG1* upregulation in response to high temperature.

## MUSHER enhances the PIR1 gene expression independently of its role in DOG1 regulation

3′RNA-seq analysis revealed 166 differentially expressed genes, including *DOG1*, between WT and *msh-1* freshly harvested seeds (Fig. 4a, Supplementary Data 3). Among the downregulated genes, *PIR1* (*PP2CA INTERACTING RING FINGER PROTEIN 1*) showed the strongest decrease (Fig. 4a). *PIR1* encodes an E3 ligase that enhances the abscisic acid (ABA) response and is highly expressed in maturing siliques[36]. Inspection of 3′RNA-seq coverage plots showed a strong reduction at both *PIR1* and an upstream localised *AT2G35320* gene (Fig. 4b). RT-qPCR confirmed the strong downregulation of *PIR1* expression in *musher* mutant, when tested in dry seeds (Fig. 4c), as well as in secondary dormancy-induced seeds (Fig. 4d), suggesting *MUSHER* activates *PIR1* expression. Consistently, both *MUSHER*-complemented

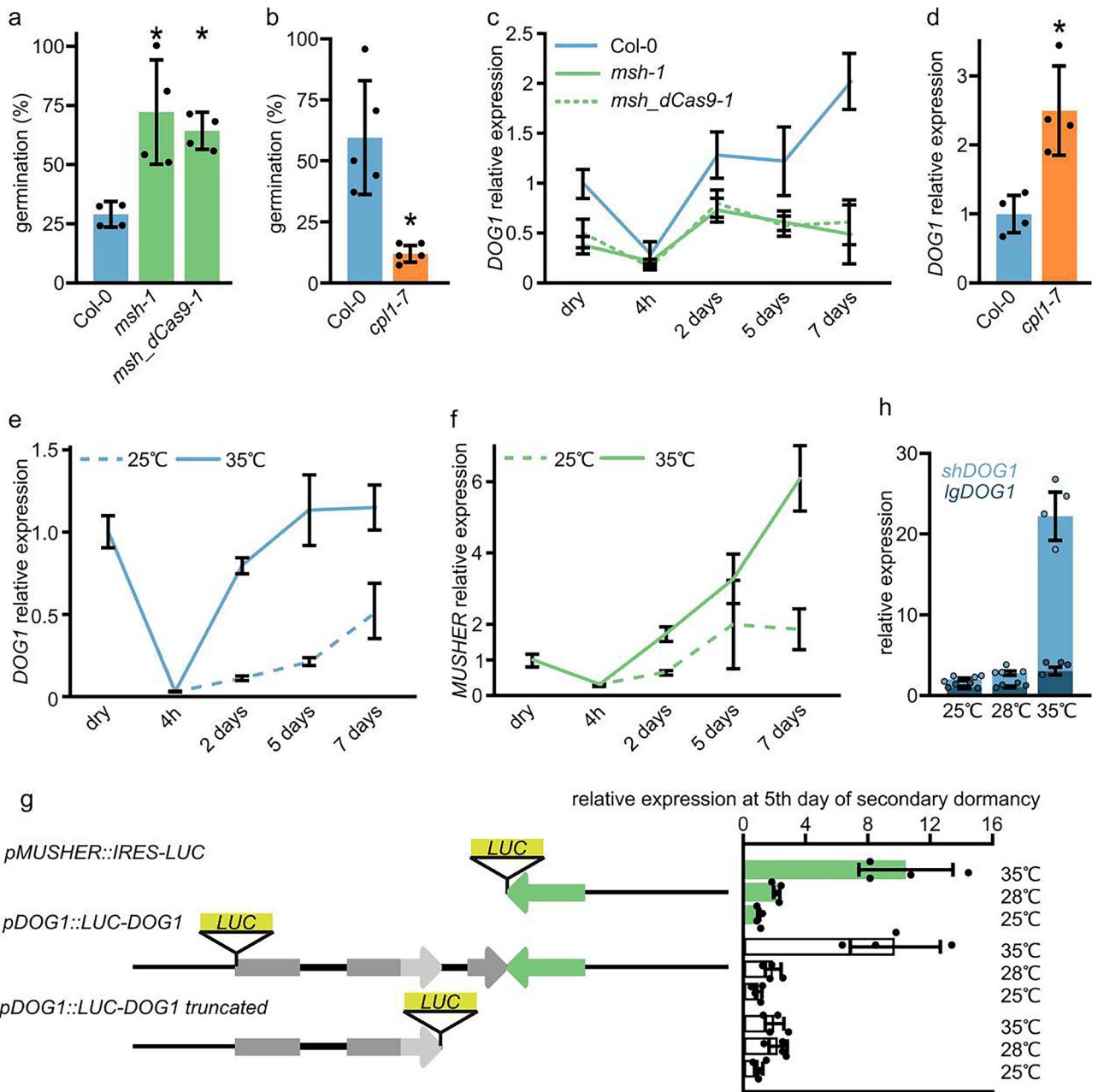

**Fig. 3 | MUSHER activates DOG1 expression during secondary dormancy induction in response to high temperature. a** Seed germination of WT and different *musher* mutants after 7 days of secondary dormancy (SD) induction at 35 °C. Germination frequency was scored after 5 days. Data are presented as mean values of four biological replicates +/− SD. *$p < 0.05$ (t-test two-sided). **b** Seed germination of WT and *cpl1-7* mutant seeds after 7 days of SD induction. Germination was scored after 5 days. Data are presented as mean values of five biological replicates +/− SD. *$p < 0.05$ (*t*-test two-sided). **c** The expression profile of *DOG1* was analysed using RT-qPCR in seeds of WT, *msh-1* and *msh_dCas9-1* subjected to secondary dormancy induction. Data are presented as mean values of four biological replicates +/− SD. *UBC21* mRNA was used as a reference gene. **d** The *DOG1* relative expression level in seeds of WT and *cpl1-7* mutant, subjected to SD induction. Data normalised to *UBC21* and related to WT. Data are presented as mean values of four biological replicates +/− SD. *p*-value from the two-tailed *t*-test, *$p < 0.05$. **e** *DOG1* and **f** *MUSHER*

expression profile in after-ripened seeds subjected to secondary dormancy induction at 25 °C and 35 °C. Expression was normalised to *UBC21* mRNA and dry seed levels. Data are presented as mean values of four biological replicates +/− SD (*$p < 0.05$, two-sided *t*-test). **g** Schematic representation of constructs introduced to transgenic plants. RT-qPCR analysis of luciferase mRNA levels in seeds subjected to secondary dormancy induction under temperatures indicated on the right. LUC expression level measured after 5 days of SD induction, normalised to *UBC21* and related to 25 °C for each construct, Data are presented as mean values of four biological replicates +/− SD (*$p < 0.05$, two-sided *t*-test). **h** Long and short *DOG1* expression analysis on the 5th day of secondary dormancy induction in WT at temperatures indicated. Data are presented as mean values of four biological replicates +/− SD (*p*-value from the two-tailed *t*-test, *$p < 0.05$). *UBC21* mRNA was used as a reference gene.

lines, *msh-1 pMUSHER_1* and *msh-1 pMUSHER_2*, showed significantly higher expression of *PIR1* than *msh-1* (Supplementary Fig. 5c). Whereas *cpl1-7* showed elevated *PIR1* levels, along with *MUSHER* overexpression (Fig. 4d). Interestingly, *PIR1* expression remained unchanged in the *dog1* mutant. This indicates that *PIR1* is activated by *MUSHER*

independently from *DOG1* (Fig. 4c,d, Supplementary Fig. 5a,b). ChIP-qPCR analysis revealed a significant decrease in Pol II occupancy throughout the entire *PIR1* locus in *msh*, suggesting that in contrast to *DOG1*, *MUSHER* regulates *PIR1* transcription levels rather than termination of *PIR1* transcripts (Fig. 4e). ChIP-qPCR analysis along *PIR1* locus

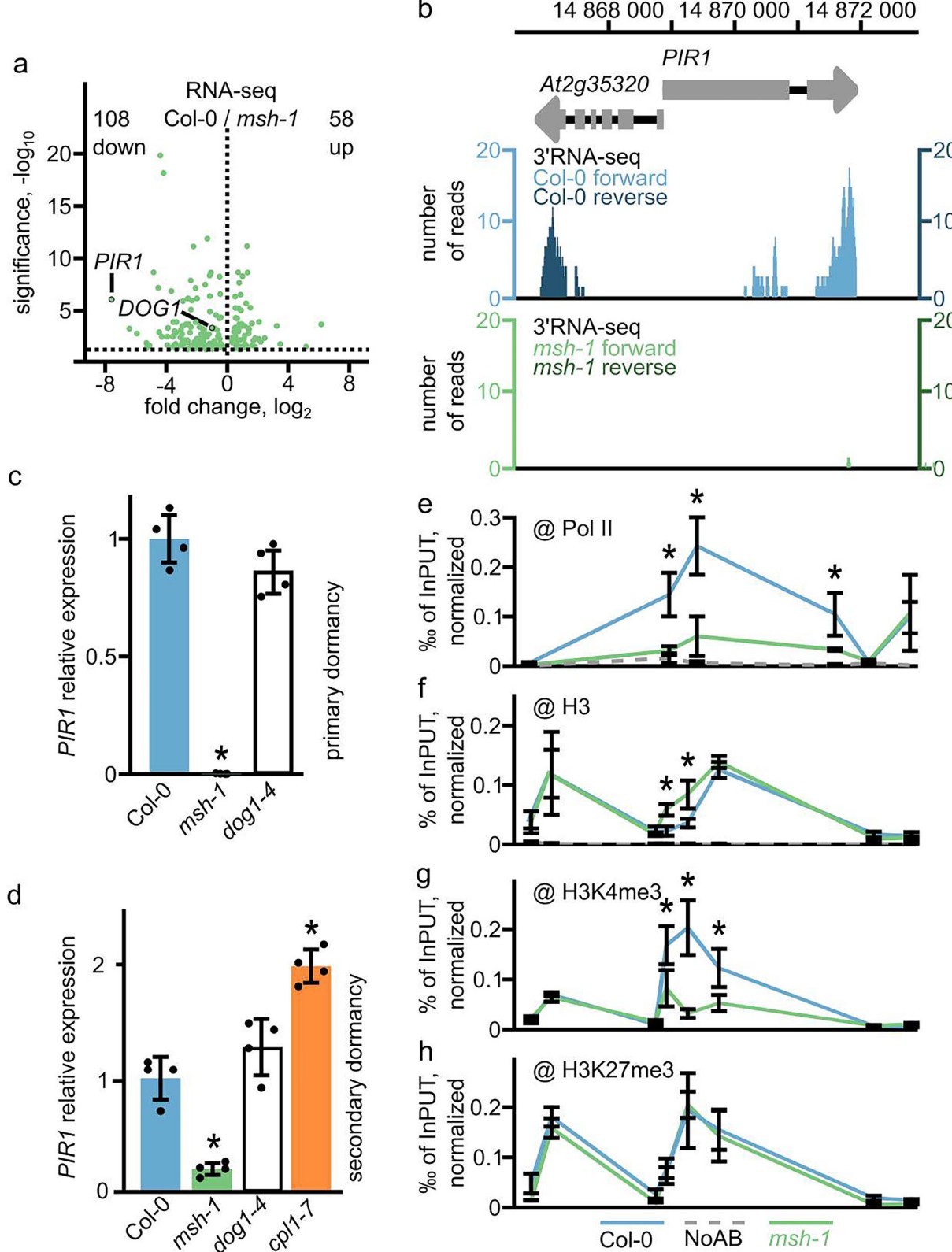

showed a TSS-localised decrease in H3K4me3, slight decrease in H3 and no change in H3K27me3 levels in *msh-1* (Fig. 4f–h), confirming the lower transcriptional activity of *PIR1* promoter.

Our results showed that *MUSHER* activates *PIR1* expression independently of its role in *DOG1* gene regulation. In addition, our data suggest that *MUSHER* regulates transcriptional activity of the *PIR1* promoter, whereas at *DOG1*, it specifically contributes to proximal

termination site selection. Importantly, both *DOG1* and *PIR1* regulation by *MUSHER* is compatible with its chromatin association (Fig. 1).

### *MUSHER* lncRNA activates *PIR1* expression through *PIR1-DOG1* chromatin interaction

We showed that *MUSHER* is chromatin-bound (Fig. 1h). To identify *MUSHER* binding sites, we conducted a Chromatin Isolation by

**Fig. 4 | MUSHER enhances PIR1 expression independently of its role in DOG1 regulation. a** Volcano plot showing differentially expressed genes for WT and *msh-1* selected using DESeq2; absolute log2FC > log2 (1.5), FDR < 0.05, *n* = 4. *DOG1* and *PIR1* genes are highlighted. The number of downregulated (down) and upregulated (up) genes is provided on the plot. **b** Diagram of 3′RNA-seq reads for *PIR1* locus in WT and *msh-1* mutant. Schematic diagram showing PIR1 locus: exons (grey rectangles), 3′UTR region (grey arrow) **c** The *PIR1* relative expression level in freshly harvested seeds (primary dormancy) of WT, *msh-1* and *dog1-4* mutants, normalised to *UBC21* and relative to WT. Data are presented as mean values of four biological replicates +/− SD (*p*-value from the two-tailed *t*-test, *\*p* < 0.05). *\*p* < 0.05. **d** The *PIR1*

relative expression level in WT *msh-1*, *dog1-4* and *cpl1-7* mutant seeds, after 5 days of SD induction, normalised to *UBC21* and relative to WT. Data are presented as mean values of four biological replicates +/− SD (*p*-value from the two-tailed *t*-test, *\*p* < 0.05). **e** Pol II **f** H3 **g** H3K4me3 and **h** H3K27me3 occupancy was measured by ChIP-qPCR in WT, *msh-1*. (*n* = 4) H3K4me3 and H3K27me3 values were normalised to H3 and to *ACTIN7* (*AT5G09810*); Pol II and H3 were normalised to *ACTIN7* (*AT5G09810*). For all ChIP experiments *\*P* < 0.05, ns - not significant (two-tailed *t*-test). Error bars represent the standard deviation of four biological replicates. H3, H3K4me3 and H3K27me3 (**f–h**) ChIP were done side by side and share NoAB control shown on panel h.

*MUSHER*-RNA Purification followed by DNA-seq (ChIRP-seq) in seeds[54]. This revealed 23 DNA regions bound by *MUSHER* lncRNA (Supplementary Data 4). The two most significantly enriched are *PIR1* and *DOG1 loci*. Quantifying the ChIRP signal at *DOG1* and *PIR1 loci* shows that target enrichment is greatly diminished in the *msh-1* mutant (Fig. 5a,b). In addition, ChIRP using *LacZ* probes, a commonly used negative control, also failed to recover significant signals at both targets (Fig. 5a,b). Analysis of the ChIRP-seq data revealed that two replicates exhibited high correlation, confirming the reproducibility of the assay (Supplementary Fig. 6a,b). In agreement with our conclusion that *MUSHER* regulates *PIR1* transcription initiation, *MUSHER* peaks centred around *PIR1* TSS. On the *DOG1* locus, *MUSHER* ChIRP-seq detected two strong peaks, one at a site where it is produced and one colocalising with the *DOG1* exon2-intron2 region, just upstream of the proximal polyA site, the selection of which is promoted by *MUSHER* (Fig. 5a).

Our data showed that the *DOG1* proximal polyA site and *PIR1* TSS are bound by *MUSHER* lncRNA. Together with the lack of *PIR1* expression changes in the *dog1* mutant (Fig. 4c and Supplementary Fig. 5b), this suggested that both *DOG1* and *PIR1* are direct and independent *MUSHER* targets. Of note, we could not find a common DNA motif or R-loop enrichment among *MUSHER* targets, indicating that *MUSHER* may not rely on a sequence-based mechanism for target specification. Whereas the *DOG1 locus* is localised next to *MUSHER* on chromosome 5, the *PIR1 locus* is localised on chromosome 2. This encouraged us to test the physical proximity of *PIR1* and *DOG1 locus* chromatin.

To this end, we used *PIR1* probes to fish out crosslinked DNA sequences followed by NGS. This assay is based on a similar principle as PiCh (Proteomics of isolated Chromatin segments)[55] but analyses DNA rather than proteins attached to a specific *locus*. This method was called chromatin-DNA precipitation assay (ChIDP). The use of glutaraldehyde combined with the covalent link between probes and beads allowed us to use more stringent conditions, resulting in a high signal-to-noise ratio (Fig. 5c,d, Supplementary Fig. 6a,c Supplementary Data 5). Examination of the ChIDP-seq results showed that two replicates were highly similar, verifying the assay's consistency (Supplementary Fig. 6a). Surprisingly, we discovered that *PIR1* DNA interacts with the proximal polyA site of *DOG1* but not with *MUSHER* (Fig. 5c,d, Supplementary Fig. 6a,d). This suggests that *MUSHER* activity on *PIR1* is mediated by long-range interaction established by the proximal polyA site of *DOG1*, which is occupied by *MUSHER*.

*DOG1* and ABA are the two main regulators of dormancy in Arabidopsis[56]. We confirmed the published results that *pir1* mutants show decreased sensitivity to ABA-mediated germination inhibition (Fig. 5e)[36]. Notably, a mutant in a gene directly upstream of *PIR1* (*at2g35320-1*) showed no significant changes in ABA sensitivity compared to WT. In agreement with published results, *dog1* mutant seed germination was suppressed by ABA in the same way as WT[27]. Importantly, multiple *musher* alleles showed much weaker sensitivity to ABA-mediated inhibition of germination across all tested ABA concentrations, similar to the *pir1* mutant (Fig. 5e, Supplementary Fig. 6e-g). Therefore, in contrast to the DOG1 protein, *MUSHER* lncRNA does modulate ABA responsiveness. Our data are consistent with a model

where *MUSHER* lncRNA enhances the ABA sensitivity by activating *PIR1* gene expression. *MUSHER's* ability to activate *DOG1* and *PIR1* points to its role in integrating the DOG1 and ABA-dependent repression of seed germination.

## U1 snRNP mediates *MUSHER's* association to chromatin

*MUSHER* physically binds to and regulates *DOG1* and *PIR1* gene expression (Figs. 1,2,5). Since most *MUSHER* transcripts are localised at the chromatin, it suggests that an active mechanism exists that sequesters *MUSHER* at the chromatin (Fig. 1h).

In addition to its well-known splicing function, U1 snRNP has been shown in mammals to bind various lncRNAs, facilitating their chromatin tethering[15]. Interestingly, we identified four putative U1 binding sites within the *MUSHER* sequence, even though we failed to detect *MUSHER* splicing (Supplementary Data 1,6 and Fig. 1c). Therefore, we analysed whether *MUSHER* interacts with U1 snRNP. Using probes specific for *MUSHER*, but not *LacZ* probes, we selectively recovered U1, but not U4 or U6 RNA using formaldehyde-assisted RNA ImmunoPrecipitation (RIP) (Fig. 6a). Supporting the specificity of the approach, U1 recovery was strongly diminished when we repeated the experiment using *MUSHER* probes in *msh-1* lines (Fig. 6b). Next, we validated the *MUSHER*-U1 association by RNA Immunoprecipitation (RIP) using the U1 snRNP component U1-70K. We used a recently reported *u1-70k* complementing line expressing GFP tagged U1-70K driven by endogenous U1-70K promoter (*pU1-70K::U1-70K-GFPu1-70k*)[21]. RT-qPCR showed that U1-70K immunoprecipitation recovered spliced control RNA, failed to recover unspliced control mRNA, and, importantly, retrieved a significant amount of *MUSHER* lncRNA (Fig. 6c).

Having shown that *MUSHER* is bound by U1 but not U6 or U4 snRNP, we asked if this contributes to *MUSHER* chromatin association. To do this, we used a recently described artificial microRNA (amiRNA) line that shows significantly downregulated *u1c* levels[21]. Subcellular localisation showed that *UBC21* and *DOG1* mRNA are nearly equally distributed between cytoplasmic, nucleoplasm and chromatin-bound fractions, as shown also in Fig. 1 (Fig. 6d). In addition, neither *UBC21* nor *DOG1* mRNA distribution among the cellular fractions was altered in the *u1c* mutant (Fig. 6d). Importantly, *MUSHER's* association to chromatin was strongly reduced in *u1c* mutant, suggesting U1 tethers *MUSHER* lncRNA at the chromatin. Next, we asked whether the U1 requirement for RNA retention at chromatin is a more general phenomenon in plants. We started by confirming that *COOLAIR*, shown to be expressed in seeds[57], is predominantly chromatin-bound, like *MUSHER* (Fig. 6d). In addition, unspliced *PUPPIES* isoform, a lncRNA involved in Pol II pausing regulation at *DOG1* in seeds[7], was also strongly enriched in chromatin fraction. Both *PUPPIES* and *COOLAIR* have multiple U1 binding sites (Supplementary Data 6) and we found that in contrast to the control mRNA, *MUSHER* and all other analysed lncRNAs showed reduced chromatin association and a corresponding increase in nucleoplasmic abundancy in the *u1c* mutant (Fig. 6d). We conclude that *MUSHER* binds U1 snRNP and that U1 facilitates *MUSHER* and other tested lncRNA-chromatin association and propose that the U1 role in chromatin retention of lncRNA is conserved between animals and plants.

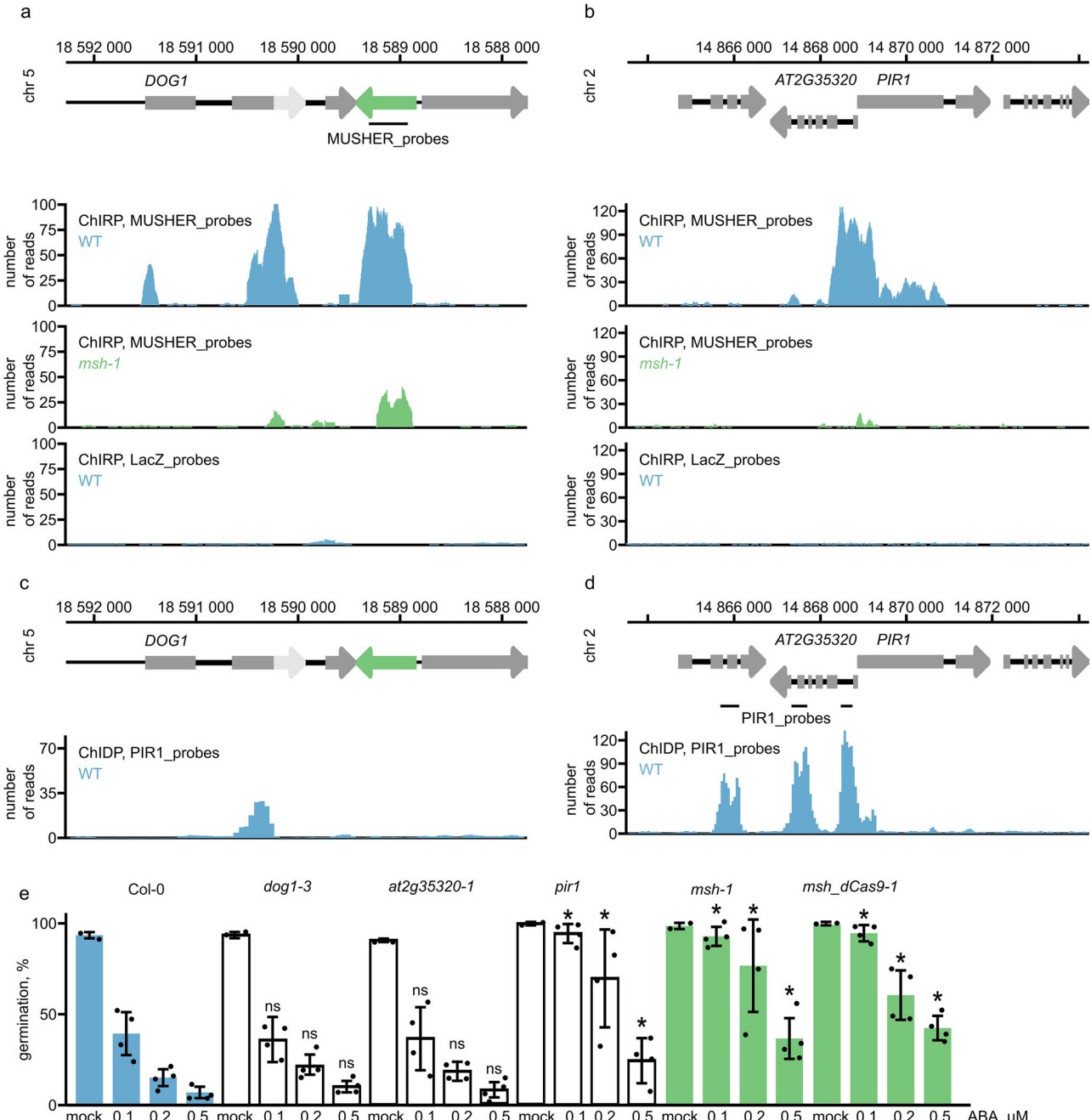

**Fig. 5 | MUSHER lncRNA integrates ABA and DOG1 pathways by interacting with PIR1 TSS and DOG1 proximal termination site.** Snapshot of ChIRP-seq in WT and *msh-1* mutant using *MUSHER*-specific and *LacZ*-specific probes on **a** *DOG1* and **b** *PIR1* loci. Schematic diagram showing *DOG1* and *PIR1* locus: exons (grey rectangles), *shDOG1* 3'UTR region (light grey arrow), *lgDOG1* 3'UTR region (grey arrow), and *MUSHER* (green). The lines below the scheme mark regions used to design *MUSHER* probes. Data are the mean of two biological replicates. **c** Snapshot of ChIDP-seq in WT using *PIR1*-specific probes on *DOG1* and **d** *PIR1* loci. Data are the

mean of two biological replicates. The lines below the scheme mark regions used to design PIR1 probes. **e** The germination rate of WT, *dog1-3*, *at2g35320-1*, *pir1* and *msh-1* mutant seeds on MS media supplemented with various ABA concentrations. Data are presented as mean values of four biological replicates for seeds germinated on ABA +/− SD and 2 biological replicates for mock. Asterix denotes a two-tailed Welch's *t*-test comparing the wild type (WT) to each mutant yielded a *p* < 0.05.

Finally, we asked if *MUSHER's* chromatin association is required for its function. Both *u1c* and another amiRNA mutant in the U1 snRNP component U1-70K, show pleiotropic phenotypes, including significant dwarfism and delayed development[21]. Thus, it would be difficult to distinguish between primary dormancy phenotype and developmental defects resulting in delayed germination. To circumvent this, we took advantage of the fact *MUSHER* also shows defects during secondary dormancy induction. To synchronise the physiological state of seed, they were matured on mother plants and stored

until they lost primary dormancy and then tested for the induced secondary dormancy. *u1c* and *u1-70k* mutant seeds showed low secondary dormancy phenotype with nearly all seeds germinating compared to less than 50% of germinated seeds in the WT (Fig. 6e, Supplementary Fig. 7). The reduction in *u1c* and *u1-70k* secondary dormancy phenotype was accompanied by decreased *shDOG1* expression (Fig. 6f). This agrees with a model where U1 is required for *MUSHER's* association to chromatin and its function in seed dormancy induction. In summary, U1 is required to tether multiple lncRNAs at the

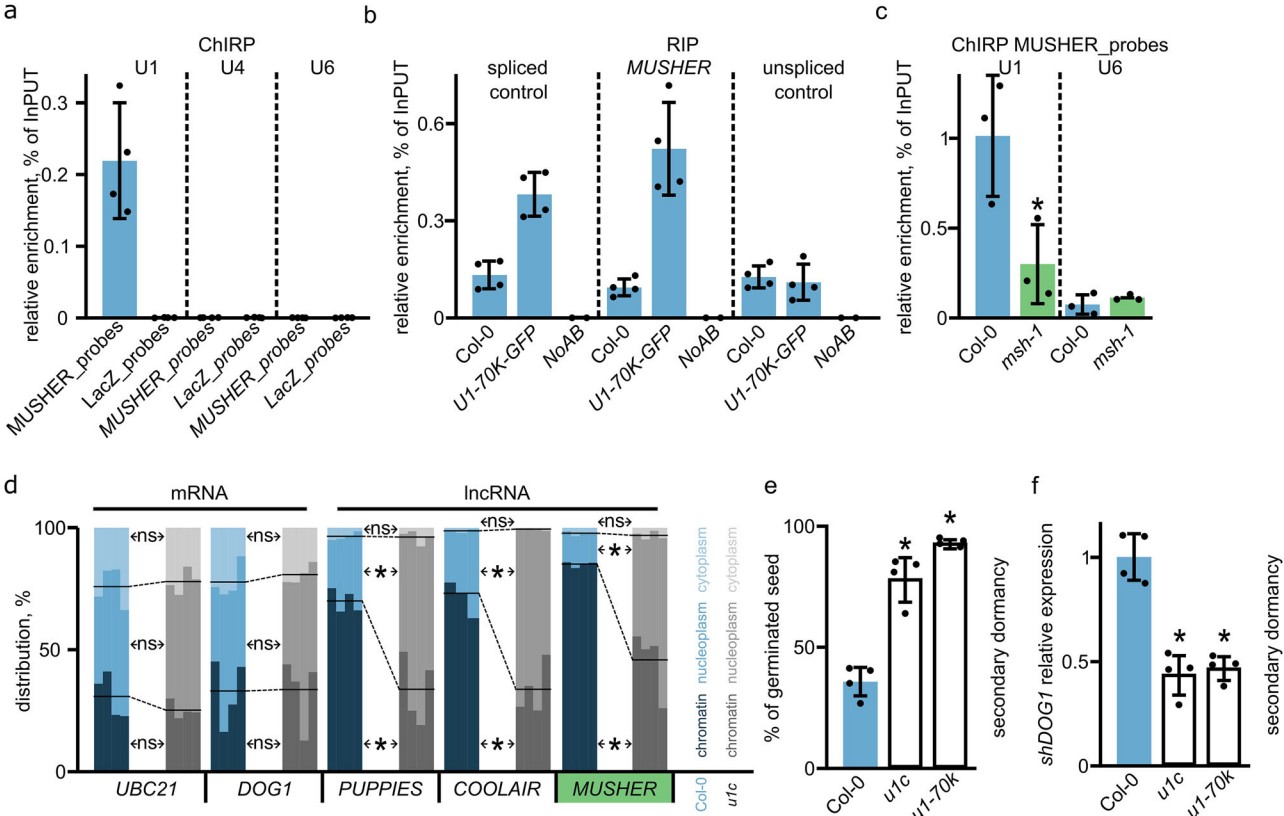

**Fig. 6 | U1 is required for chromatin tethering of MUSHER and other lncRNA. a** *MUSHER* or *LacZ* probes were used in formaldehyde-assisted RIP to enrich associated RNA. U1, U4 and U6 recovery was analysed using RT-qPCR. Data are the mean of four biological replicates +/− SD. **b** *MUSHER* probes were used to enrich *MUSHER* from WT and as a negative control in *msh-1*. U1 and U6 recovery was analysed using RT-qPCR. The data represent the average of four biological replicates +/− SD. **c** U1A protein was precipitated side by side with a no-antibodies (NoAB) control from plants expressing *pU1-70K::U1-70K-GFP u1-70k*, and co-precipitated RNA was analysed using RT-qPCR for spliced (*AT5G45830*) and unspliced (*AT5G45820*) RNA controls and *MUSHER*. The data are averaged from three biological replicates +/− SD. A two-tailed Welch's *t*-test *p*-value was calculated compared to WT (*$p < 0.05$). **d** Subcellular localisation of mRNAs and lncRNAs in maturing Arabidopsis seeds of WT and *u1c* amiRNA line. Lines represent the mean of chromatin and nucleoplasmic fractions contribution of the total amount of indicated transcript. The blue-shaded colour refers to Col-0, while the grey-shaded colour corresponds to *u1c*. Each transcript was measured in 4 replicates represented as a separate bar. *$p < 0.01$ from two-tailed Student *t*-test. **e** After ripened seeds of WT (WT), *u1c* amiRNA and *u1k-70k* mutants were induced into secondary dormancy, moved to permissive conditions and scored for germination ability. Germination (%) was scored as radical protrusion. Data are the mean of four biological replicates +/− SD (two-tailed *t*-test, *$p < 0.05$). **f** *shDOG1* expression was analysed in WT, *u1c* amiRNA and *u1-70k* mutants seeds induced into secondary dormancy. Data are the mean of four biological replicates +/− SD. A two-tailed *t*-test *p* was calculated compared to WT (*$p < 0.05$). Normalised to *UBC21* and relative to WT.

chromatin, and in case of *MUSHER*, we showed this chromatin association is required for the lncRNA function.

## Discussion

Our results indicate that *MUSHER* modulates seed dormancy through the transcriptional regulation of DOG1 and ABA sensitivity pathways (Fig. 7a). We show that *MUSHER* lncRNA is chromatin-bound and requires U1 snRNP for its retention and activity at chromatin (Fig. 7b).

### *MUSHER*-mediated regulation of *DOG1* and *PIR1* expression

Our analysis showed that *MUSHER* employs two different mechanisms to activate *DOG1* and *PIR1*. *MUSHER* promotes *DOG1* expression by enhancing CPSF complex recruitment to the proximal cleavage and polyadenylation site, resulting in enhanced production of a functional short *DOG1* alternative polyadenylation isoform (Fig. 2). In agreement, the *MUSHER* effect on *DOG1* is not accompanied by changes in Pol II recruitment or H3K4me3 levels at the *DOG1* gene (Fig. 2 and Supplementary Fig. 3). In contrast, *MUSHER* activates *PIR1* expression by enhancing Pol II recruitment and consequent H3K4me3 deposition at the *PIR1* promoter (Fig. 4). Similar to *HID1* lncRNA[13], and differently then *COOLAIR*[58] and *APOLO*[5], *MUSHER* promotes H3K4me3 deposition without affecting H3K27me3 repressive marks. Notably, *HID1*

represses transcription by blocking the binding of ARABIDOPSIS TRITHORAX-RELATED7 (ATXR7), an H3K4me3 methyltransferase, to nucleosomes, leading to increased H3K4me3 levels when HID1 is inactivated[13]. In contrast, *MUSHER* inactivation reduces H3K4me3 deposition at the *PIR1* promoter, suggesting that *MUSHER* and *HID1* regulate H3K4me3 through distinct molecular mechanisms. Of note, we did not detect *PIR1* proximal polyadenylation either in our seed-specific 3'RNA-seq data (Fig. 4b) or in the published long read-based transcript mapping dataset[39]. It is currently not clear why *MUSHER*-dependent *DOG1* and *PIR1* regulation are based on different modes of activity. It is possible that *MUSHER* acts in the same way on *DOG1* and *PIR1*, but the local chromatin environment of the target or *MUSHER* site of binding with respect to the target transcription start site results in seemingly different mechanisms. For example, the recruitment of CPSF complex to the gene promoter may enhance transcription initiation or transition to efficient elongation. We, however, favour a model where *MUSHER* employs two different mechanisms to enhance *DOG1* and *PIR1* expression. This view is supported by the fact that *DOG1* expression is reduced roughly by 50%, while *PIR1* expression is undetectable in multiple *msh* mutants tested.

Chromatin precipitation using *MUSHER* RNA (ChIRP) followed by sequencing revealed only a very limited number of *MUSHER* targets,

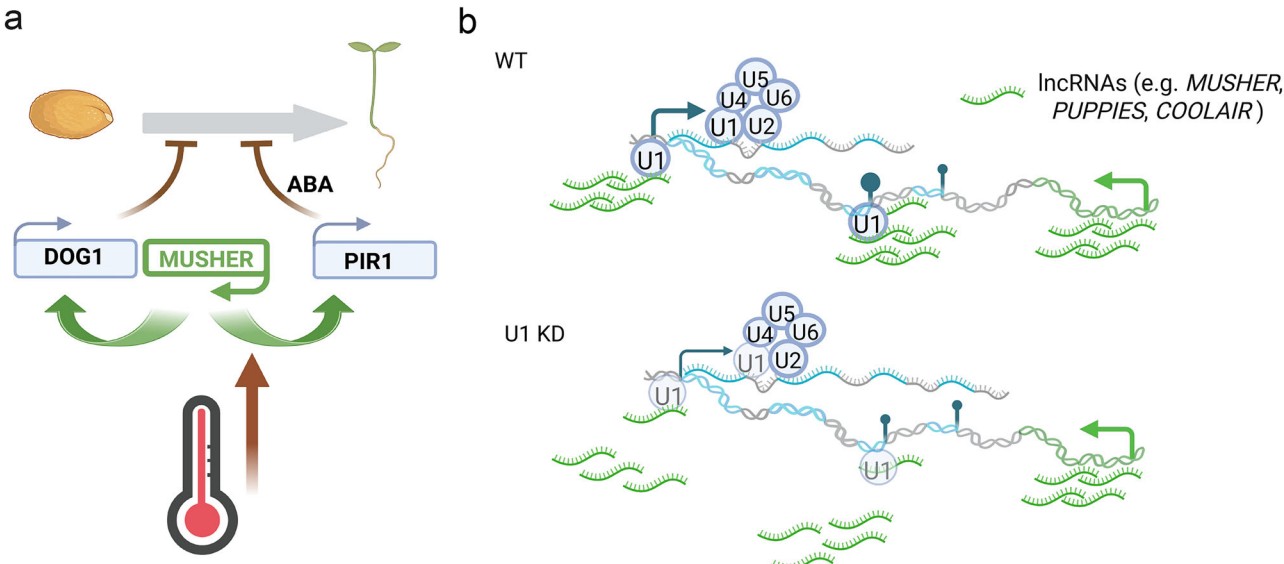

**Fig. 7 | MUSHER regulates seed dormancy and lncRNA chromatin retention in Arabidopsis. a** MUSHER modulates seed dormancy through the transcriptional regulation of *DOG1* and *ABA* pathways. Created in BioRender Halale manjunath, V. (2025) https://BioRender.com/djj6tmh. **b** U1 is required for chromatin retention of lncRNA, including *MUSHER* in Arabidopsis. Created in BioRender. Halale manjunath, V. (2025) https://BioRender.com/zss2438.

with *DOG1* and *PIR1* being the most significantly enriched. The high selectivity of *MUSHER* is somehow surprising, as many of the published plant lncRNAs show binding to multiple sites spread across the genome[5,59]. *PIR1* DNA chromatin-DNA precipitation assay (ChIDP) followed by sequencing showed that *PIR1* DNA also has a very limited number of DNA targets, with, excluding *PIR1* bait, *DOG1* being the strongest. This suggests that high *MUSHER* selectivity could be guided by a spatial arrangement of DNA in seeds.

### MUSHER as a Link Between ABA Sensitivity and DOG1

Interestingly, both *MUSHER* targets encode proteins, PIR1 and DOG1, that interact with PP2CA protein, a regulator of ABA signalling[35]. PIR1 binding to PP2CA promotes its degradation, while DOG1 inhibits PP2CA phosphatase activity[35,37]. Both *dog1* and ABA synthesis deficient *nced2/5/6/9* mutants fail to enter dormancy. Simultaneously, *dog1* does not show major changes in ABA levels or sensitivity[27,60] and *nced2/5/6/9* shows unaffected *DOG1* induction kinetics during secondary dormancy establishment[53]. This suggests that DOG1 and ABA synthesis pathways function independently in dormancy regulation. We demonstrate that *MUSHER* inactivation leads to decreased transcription of *DOG1* and *PIR1*, which results in decreased ABA sensitivity. We propose that *MUSHER* lncRNA facilitates the concurrent activation of DOG1 and ABA pathways, integrating them into a shared regulatory network.

High temperatures are known to enhance *DOG1* expression and ABA biosynthesis in seeds[51,61]. We observed altered ABA sensitivity and reduced secondary dormancy induction in response to high temperature in *msh* mutants. Elevated temperatures contribute to two well-described responses: (1) thermoinhibition, a high-temperature-mediated inhibition of germination in light[62] and (2) secondary dormancy induction in non-dormant imbibed seeds exposed to high temperatures in the darkness[31,49,51]. We hypothesise that *MUSHER*-mediated integration of ABA and DOG1 pathways contributes not only to dormancy but also to other ABA/DOG1-related phenotypes (e.g., drought tolerance or thermoinhibition). We conclude that, similar to other long non-coding RNAs involved in temperature sensing, such as *COOLAIR*[8], *COLDWRAP*[63] and *SVALKA*[6] in plants, and *Hsrω*[64] and *Heat*[65] in animals, *MUSHER* also plays a role in temperature sensing in plants.

### U1 role in chromatin retention of lncRNA in Arabidopsis

In contrast to mature mRNAs, lncRNAs are often enriched in the nucleus[1,15]. Consistently, many lncRNA have been shown to regulate gene expression by binding to chromatin[1,5,6,39]. The molecular mechanism of human lncRNA retention in the nucleus has only recently started to be explored[15,66,67], and remains largely unknown in plants. Mechanisms ensuring human lncRNA nuclear retention encompass the presence of specific sequences[68], including *Alu* transposable elements binding the HNRNPK complex[69]. These protein-RNA interactions may contribute to liquid-liquid phase separation (LLPS) that can also be responsible for the nuclear retention of lncRNA. For example, LLPS by *NEAT1* lncRNA results in the formation of nuclear paraspeckles[70] and *XIST* lncRNA was reported to undergo LLPS with different partners localised at inactive X chromosome[71]. However, nuclear retention may not be enough to guide lncRNA directly to chromatin. This has been proposed to be a result of direct RNA-DNA interaction, either in the form of R-loops or RNA:DNA triplex[15]. In addition, U1 snRNP, involved in 5′ splice site recognition, has recently been shown to be responsible for the chromatin binding of a large fraction of lncRNAs[15]. In contrast to *Alu* and other sequence elements identified in humans, U1 binding sites are well conserved in plant genomes.

Plant lncRNA retention at chromatin has not been explored, but RNA-seq analysis indicates that lncRNAs are enriched in the nucleus[72]. Also, like their human counterparts, plant lncRNAs often regulate gene expression by chromatin binding[6], indicating that their chromatin retention may be actively maintained. Multiple plant lncRNAs have been described, but few of them have been reported to be expressed in seeds and to control gene expression by chromatin-related mechanisms. This includes *COOLAIR* regulating the *FLC* gene chromatin state[9,73], *PUPPIES* regulating Pol II pausing at the *DOG1* gene[7] and, described here, *MUSHER* promoting the *DOG1* gene proximal cleavage and polyadenylation site selection. In seedlings, *COOLAIR* is retained at chromatin, presumably by its ability to form R-loops[74] and a cloud around its transcription site[10]. Similarly, *APOLO* lncRNA has been shown to form R-loops at its targets located in *trans* to its transcription site[5]. Here, we propose that U1-mediated chromatin binding of lncRNAs is a largely sequence-independent mechanism used by plants

to retain lncRNA at chromatin. We showed that U1 snRNP is required for *MUSHER*, *COOLAIR* and *PUPPIES* to bind the chromatin efficiently. As we did not test the interplay between U1 binding and LLPS, we cannot exclude the possibility that U1 binding is required or otherwise interlinked with the phase separation of lncRNA at chromatin. R-loops have not been reported in seeds, and the one shown in seedlings at *PIR1* locus[75] is unlikely to be formed by *MUSHER* since *MUSHER* is very lowly expressed in seedlings. Therefore, while we cannot exclude this possibility, *MUSHER* is unlikely to form R-loops. As concluded for humans, U1 lncRNA targets[15] do not form direct R-loops with the target DNA. This suggests that in plants, R-loops probably do not contribute to U1-mediated lncRNA retention. The fact that the lncRNA tested here have different modes of action, implies that the U1 role in chromatin retention is not limited to a specific lncRNA activity. Altogether, our data strongly supports the idea that the U1 role in chromatin retention of lncRNA is evolutionarily conserved between animals and plants.

## Methods

### Plant material and growth conditions
WT used in this work is Columbia (Col-0) *Arabidopsis thaliana* accession. Plants were grown on soil in a greenhouse under long-day conditions (16 h light/8 h dark) at 22 °C/18 °C. All mutants and transgenic lines were created in Col-0 background and were homozygous for the indicated genotype. *MUSHER* lines *msh-1*(SM_29297) and *msh-2*(SM_29310) were obtained from the NASC centre. Other mutants used in this study have been described previously: *cpl1-7*[76], *cpl1-8* (GK-165H09)[32], *cpl1-9* (GK-849H10)[32], *dog1-3* (SALK_000867)[27], *dog1-4* (SM_3_20886)[30], *fy-2* (SAIL_ 657D4)[77], *esp1-2* (SALK_078793)[78], *pir1* (SALK_078237)[36], *amiR-u1-70k* and *amiR-u1-c*[21].

### Germination tests
Germination assays were performed as described previously to estimate primary dormancy levels[7,30]. Briefly, 100–200 freshly harvested seeds were sown on Petri plates with blue germination paper (HoffmanMfg) and a layer of water-soaked fabric underneath to keep moisture. The plates were sealed and placed in the Percival chamber under long-day conditions (16 h light/8 h dark) at 22 °C/18 °C. Photographic documentation was performed once a day, and seeds with root protrusion were counted as germinated.

To evaluate secondary dormancy induction levels, seeds were sown on Petri plates with blue germination paper (HoffmanMfg) and a layer of water-soaked fabric underneath to keep moisture. The plates were sealed and incubated in the darkness at varying temperatures (25 °C, 28 °C, 35 °C) for 7 days. Subsequently, plates were transferred to the Percival chamber under long-day conditions (16 h light/8 h dark) at 22 °C/18 °C.

An abscisic acid (ABA) sensitivity test (Fig. S5) was conducted using the genotypes Col-0, *dCas9-control-1*, *msh-1*, *msh-2*, *msh_dCas9-1*, *msh_dCas9-2*, *pir1*, *dog1-4* and *msh-1xpir1*. Seeds were sown on blue paper soaked with 45 ml water supplemented with 0.1 μM ABA and stratified for 24 h before transferring to 22 °C (long day conditions). Differences in germination rates across genotypes were measured after 4 days and assessed using a Kruskal–Wallis test within each treatment, followed by Dunn's post-hoc test for pairwise comparisons.

### Cloning, plant transformation and CRISPR-dCas9 mutant generation
The generation of the dCas9 transgenic plants was performed according to Čermák et al.[42]. The specific sgRNAs to Arabidopsis TAIR10 genome were designed using the CRISPR-P 2.0 (http://crispr.hzau.edu.cn/cgi-bin/CRISPR2/CRISPR) to select the canonical NGG PAM motif. sgRNAs were cloned to the dCas9 vector by Golden Gate assembly (*dCas9_control-1*, *dCas9_control-2*, *msh_dCas9-1* and *msh_dCas9-2*) according to the protocols 3B and 5[42]. Primers used for cloning are listed in Supplementary Data 7. All constructs were verified

by Sanger sequencing. Each construct was transformed into Agrobacterium strain GV3101 and then transformed into Col-0 or an appropriate mutant background using the floral dip method. T1 transformant plants were identified by selecting the resistance to glufosinate. The seeds from the selected T1 plants were used for phenotypic analysis. At least two independent transgenic lines were tested and propagated. As a control for potential indirect effects from dCas9 expression, we also created two dCas9 lines with guides recognising different intergenic regions localised 100 kb from *DOG1* locus: *dCas9_control_1* and *dCas9_control_2*, which showed no influence on *MUSHER* transcription. To generate *MUSHER*-complemented lines, *msh-1 pMUSHER_1* and *msh-1 pMUSHER_2*, we modified the *pGWB635_LUC* vector by introducing the *MUSHER* genomic sequence along with its promoter region (>4 kb). Although both constructs share the same *MUSHER* promoter start site in the *Arabidopsis thaliana* genome, the luciferase (LUC) reporter integration begins at position 18,589,360 for *pMUSHER_1* and 18,589,685 for *pMUSHER_2*. Due to the shared BASTA resistance in *msh-1* and the *pGWB635_LUC* plasmid, we crossed *pMUSHER_1* in the Col-0 background with *msh-1* plants to generate complemented lines, selecting double-homozygous individuals in subsequent generations.

### Luciferase measurement
100 seeds placed on a microscope glass slide were squashed to release the embryo from the seed coat. After adding D-Luciferin (Beetle luciferin potassium salt, Promega) and 20 min of incubation in darkness, the light signal was measured using a NightSHADE camera (Berthold Technologies GmbH & Co.KG) with a 10-minute exposure.

### RNA extraction
RNA extraction protocol was adapted from Kowalczyk et al.[32]. After grinding in liquid nitrogen, samples were resuspended in 500 μl of homogenisation buffer (100 mM Tris pH 8.5, 5 mM EDTA, 100 mM NaCl, 0.5% SDS, 1% β-mercaptoethanol) and centrifuged for 5 min at room temperature at 8000 g. The supernatant was transferred to the new tubes containing 250 μl of chloroform. After 15 min of shaking, 250 μl of phenol was added, and shaking was repeated, followed by centrifugation for 10 min at 10,000 g. The upper phase was transferred to new tubes, and an equal volume of phenol:chloroform:isoamyl alcohol was added. Shaking and centrifugation steps were repeated. The upper phase was mixed with 10% of the volume of 3 M sodium acetate (pH 5.2) and an equal volume of isopropanol. After centrifugation for 30 min at 20,000 g at 4 °C, the pellet was washed with 70% EtOH and resuspended in water.

### RT−qPCR
Trace DNA in the RNA samples was removed using a TURBO DNA-free kit (Thermo Fisher Scientific) according to the manufacturer's protocol. The DNA removal was confirmed by PCR using primers to the *PP2A* gene (AT1G13320). The RNA sample quality and quantity were checked by agarose gel electrophoresis and using a Nanodrop 2000 spectrophotometer. Reverse transcription using 2.5 μg of RNA was performed with a Superscript III cDNA Synthesis kit (Thermo Fisher Scientific) according to the manufacturer's protocol. An equal mixture of oligo(dT) and hexamer primers was used for cDNA synthesis according to the manufacturer's protocol. 5x diluted cDNA was used as a template in the qPCR reaction (LightCycler 480; Roche) with SYBR Green mix (Roche). The relative transcript level was determined with the $2 - \Delta\Delta Ct$ method[28] and normalised to mRNA of the *UBC21* (AT5G25760) gene. The primers used are listed in Supplementary Data 7.

### Chromatin ImmunoPrecipitation (ChIP)
ChIP was performed as described[32] with the following modifications. 100 mg of fresh siliques (13−20 days after pollination) were ground in

liquid nitrogen and resuspended in 20 ml of crosslink buffer (10 mM HEPES-KOH pH 7.4, 50 mM NaCl, 100 mM sucrose, 1% formaldehyde). After 10 min incubation on a rotator at 4 °C, the crosslinking agent was quenched by adding 1.25 ml of 2 M glycine. After another 5 min incubation on a rotator, samples were centrifuged at 4 °C and 2000 g for 15 min. The pellets were resuspended in 20 ml of cold Honda Buffer (20 mM HEPES-KOH pH 7.4, 0.44 M sucrose, 1.25% Ficoll, 2.5% Dextran T40, 10 mM MgCl$_2$, 5 mM DTT, 0.5% Triton X-100, 10 mM β-mercaptoethanol, 1 mM PMSF, 1x Complete protease inhibitors (Roche)) and filtered through Miracloth. Samples were centrifuged for 15 min at 2000 g at 4 °C. The pellet was resuspended in ChIP lysis buffer (50 mM HEPES-KOH pH 7.4, 10 mM EDTA, 1% SDS, 1x Complete protease inhibitors (Roche)) and sonicated using Bioruptor (Diagenode) to fragments of 300–500 bp. Samples were next diluted 10 times with ChIP Dilution Buffer (1.1% Triton X-100, 1 mM EDTA, 16.7 mM HEPES-KOH pH 7.4, 167 mM NaCl, 1 mM DTT, 1 mM PMSF, 1x Complete protease inhibitors (Roche)). Then antibodies against RNA polymerase II subunit B1 (Agrisera, AS11 1804), H3 (Abcam, ab1791), H3K4me3 (Millipore, 07-473), H3K27me3 (Abcam, ab6002) or CPSF73 (PhytoAB, PHY1043A) were added followed by addition of the mixture of Dynabeads Protein A and Protein G beads (Thermo Fisher Scientific). Next, samples were incubated overnight on a rotator at 4 °C. Two washing steps with low salt buffer (150 mM NaCl, 1 mM EDTA, 10 mM HEPES-KOH pH 7.4, 0.1% Triton X-100) and two washing steps using high salt buffer (500 m M NaCl, 1 mM EDTA, 10 mM HEPES-KOH pH 7.4, 0.1% Triton X-100) were performed. DNA was released from beads and reverse-crosslinked by incubation at 95 °C with 10% Chelex 100 suspension for 15 min. After Proteinase K treatment (55 °C) for 90 min and repeated incubation at 95 °C for 10 min, samples were centrifuged for 5 min at RT at 10,000 g. Recovered DNA was used in qPCR reaction with primers listed in Supplementary Data 7.

## RNA ImmunoPrecipitation (RIP)

For U1-70K-GFP RIP, we used 100 mg of fresh siliques (13–20 days after pollination), which were ground in liquid nitrogen and resuspended in 20 ml of crosslink buffer (10 mM HEPES-KOH pH 7.4, 50 mM NaCl, 100 mM sucrose, 1% formaldehyde). After 10 min incubation on a rotator at 4 °C, the crosslinking agent was quenched by adding 1.25 ml 2 M glycine. After another 5 min incubation on a rotator, samples were centrifuged at 4 °C and 2,000 g for 15 min. The pellets were resuspended in 20 ml of cold Honda Buffer (20 mM HEPES-KOH pH 7.4, 0.44 M sucrose, 1.25% Ficoll, 2.5% Dextran T40, 10 mM MgCl$_2$, 5 mM DTT, 0.5% Triton X-100, 10 mM β-mercaptoethanol, 1 mM PMSF, 1x Complete protease inhibitors (Roche), and 5 U/ml murine RNase inhibitor (NEB)) and filtered through Miracloth. Samples were centrifuged for 15 min at 2000 g at 4 °C. The pellet was resuspended in Nuclear Lysis Buffer (50 mM HEPES-KOH pH 7.4, 10 mM MgCl$_2$, 1% Triton X-100, 1x Complete protease inhibitors (Roche), 20 U/ml murine RNase inhibitor (NEB)), followed by 20 min incubation at 37 °C with TURBO DNASe (Thermo Fisher Scientific). After adding SDS up to 1%, samples were sonicated using Bioruptor (Diagenode). Next, samples were diluted 10 times with RIP Dilution Buffer (0.5 mM EDTA, 10 mM HEPES-KOH pH 7.4, 150 mM NaCl, 20 U/ml murine RNase inhibitor (NEB)) and incubated with GFP Agarose beads (Chromotek) for 1 h on a rotator at 4 °C. After three washing steps with RIP Wash Buffer (150 mM NaCl, 1 mM EDTA, 10 mM HEPES-KOH pH 7.4, 0.05% NP-40, 20 U/ml murine RNase inhibitor (NEB)), RNA was eluted (150 mM NaCl, 1 mM EDTA, 10 mM HEPES-KOH pH 7.4, 1% SDS, 20 U/ml murine RNase inhibitor (NEB)) and reverse-crosslinked by incubation at 95 °C for 15 min. After 1 h Proteinase K treatment (55 °C), RNA was isolated with Trisure (Bioline) according to producer protocol. After cDNA synthesis with SuperScript III (Thermo Fisher Scientific), samples were used in qPCR reaction with appropriate primers listed in Supplementary Data 7.

## Nuclei purification for ChIRP, RIRP and ChIDP

To uncover RNA-RNA (RNA Isolation by RNA Purification), RNA-DNA (Chromatin Isolation by RNA Purification) and DNA-DNA (Chromatin Isolation by DNA Purification Assay) interactions, we used 400 mg of fresh siliques (13–20 days after pollination), which were ground in liquid nitrogen and resuspended in 20 ml of crosslink buffer (10 mM HEPES-KOH pH 7.4, 50 mM NaCl, 100 mM sucrose, 1% formaldehyde (for RNA-RNA and RNA-DNA) or 0.5% glutaraldehyde (for ChIDP)). After 10 min incubation on a rotator at 4 °C, the crosslinking reaction was quenched by adding 1.25 ml 2 M glycine. After an additional 5 min incubation on a rotator, samples were centrifuged at 4 °C and 2,000 g for 15 min. The pellets were resuspended in 20 ml of cold Honda Buffer (20 mM HEPES-KOH pH 7.4, 0.44 M sucrose, 1.25% Ficoll, 2.5% Dextran T40, 10 mM MgCl$_2$, 5 mM DTT, 0.5% Triton X-100, 10 mM β-mercaptoethanol, 1 mM PMSF, 1x Complete protease inhibitors (Roche) and 5 U/ml murine RNase inhibitor (NEB)) and filtered through Miracloth. Subsequently, samples were centrifuged for 15 min at 2000 g at 4 °C. According to downstream assays, the nuclei pellets were resuspended in appropriate buffers (Element Zero Biolabs kit).

## Chromatin isolation by RNA purification assay (ChIRP)

To enrich specific RNA molecules or chromatin fragments, we used MagIC beads with custom probes targeting those sequences (Element Zero Biolabs). Element Ziololabs designed probes are delivered with kits and remain confidential as protected intellectual property.

For *MUSHER* probes RIRP and ChIRP with *MUSHER* and *lacZ* probes, nuclei pellets were resuspended in 1.1 ml of MagIC Lysis Buffer FA (Element Zero Biolabs kit), supplemented with 1x Complete protease inhibitors (Roche) and 5 mM DTT. Next, samples were sonicated using a Bioruptor (Diagenode) and diluted two times with MagIC Lysate Dilution Buffer FA, with 1% of the volume saved as an input sample. MagIC beads were pre-cleared according to the producers' protocol and added to the samples that were next incubated at 40 °C for 30 min with occasional shaking. This was followed by three washes using MagIC Wash Buffer I FA and nucleic acids elution by incubation at 92 °C for 5 min with 10 mM Tris-HCl pH 7.5. After Proteinase K treatment (55 °C, 90 min) and phenol:chloroform:isoamyl alcohol extraction, DNA and RNA were precipitated by adding 1 μl glycogen, 10% of the volume of 3 M LiCl, and an equal volume of isopropanol. Samples were centrifuged for 30 min at 4 °C at 20,000 g. Pellets were washed with cold 80% ethanol, resuspended in water, and samples were split in half. The presence of specific DNA fragments interacting directly or indirectly with *MUSHER* probes was validated by qPCR analysis. To identify RNAs enriched with *MUSHER* probes, half of the sample was incubated with TURBO DNASe (Thermo Fisher Scientific), extracted with phenol:chloroform: isoamyl alcohol as above, and used for cDNA synthesis and qPCR analysis. For ChIRP library preparation, DNA fragments were eluted from MagIC beads with RNase If (NEB). Libraries were prepared with the NEBNext® Ultra™ DNA Library Prep Kit for Illumina using NEBNext Multiplex Dual Index Primers. Paired-end sequencing was performed on the Illumina NovaSeq 6000.

## Chromatin isolation by DNA purification assay (ChIDP)

For ChIDP, the nuclei pellet was resuspended in 1.1 ml of MagIC Lysis Buffer FA (Element Zero kit) supplemented with 1x Complete protease inhibitors (Roche) and 5 mM DTT. Samples were sonicated using Bioruptor (Diagenode), diluted with MagIC Lysate Dilution Buffer FA, and split into halves (treated with RNaseA at 37 °C for 20 min and non-treated). MagIC beads recognising the *PIR1 locus* were mixed with the lysates that were next incubated at 81 °C for 5 min with occasional shaking to denature DNA partially. Subsequently, samples were transferred to 40 °C for 20 min to enable binding of the probes. This was followed by six washing steps with the MagIC Lysate Wash buffer I and II, and DNA was eluted with 10 mM Tris-HCl pH 7.5 at 92 °C for

5 min. After adding an equal volume of MagIC Lysate Wash buffer I to eluted samples, proteinase K treatment (55 °C, 120 min) was performed. To extract DNA fragments, an equal volume of phenol:chloroform:isoamyl alcohol was added. The upper phase was mixed with 1 μl of glycogen, 10% of the volume of 3 M LiCl and an equal amount of isopropanol. After centrifugation for 30 min at 20000 g at 4 °C, the pellet was washed with 80% EtOH and resuspended in water. Libraries were prepared with the NEBNext® Ultra™ DNA Library Prep Kit for Illumina using NEBNext Multiplex Dual Index Primers. Paired-end sequencing was performed on the Illumina NovaSeq 6000.

## ChIRP-seq and ChIDP-seq analysis

For ChIRP-seq and ChIDP-seq data analysis, Bowtie2 (v2.5.3)[79] was used to align the reads to the reference genome (TAIR10, with default parameters). To identify significantly enriched regions, MACS3 was used with the default parameters: bandwidth:300; model fold of 5:50; q-value cutoff: 0.05 (v3.0.0a6)[80]. bamCoverage (v3.5.4)[81] with CPM normalisation was used to create bigwig files.

**5'RACE-seq.** Analysis was performed as described previously[7]. Shortly, 500 ng of total RNA, after DNAse treatment, was used as the template for cDNA generation using SuperScript II (Thermo Fisher Scientific). The resulting cDNA was purified using AMPure XP magnetic beads (Beckman Coulter) and amplified in a series of three PCR reactions with Phusion polymerase and specific primers (1st PCR: only TSO_n1, 2nd PCR: TSO_n2 and MUSHER_5RACE, 3rd PCR: Illumina indexing primers). The final PCR product was sequenced using Illumina MiSeq. For data analysis, STAR (v2.7.8a)[82] was used to align the reads to the reference genome, followed by UMI-based PCR duplicates removal using UMI-tools (v1.1.0)[83]. The position of the first 5' nucleotide was extracted using bedtools (v2.30.0)[84].

**3'RACE-seq.** Analysis was performed as described previously[7] and based on the earlier procedure[85]. The ligation of the pre-adenylated adaptor to the 3'end of the RNA was performed using truncated T4 RNA Ligase 2 (NEB). RNA ligated with RA3_15N adaptor (containing UMI) was cleaned with AMPure XP magnetic beads and subjected to reverse transcription reaction with SuperScript III (Thermo Fisher Scientific). After three rounds of PCR with specific primers (1st PCR: MUSHER_3RACE and RTPXT, 2nd PCR: mXTf and mXTr, 3rd PCR: Illumina indexing primers), each followed with AMPure beads purification, prepared libraries were sequenced using Illumina MiSeq. For data analysis, STAR (v2.7.8a)[82] was used to align the reads to the reference genome, followed by UMI-based PCR duplicates removal using UMI-tools (v1.1.0)[83]. The position of the last 3' nucleotide was extracted using bedtools (v2.30.0).

**3'RNA-seq.** Analysis was performed as described previously[49]. Briefly, 0.5 μg of total RNA was reverse transcribed with UMI-containing oligo(dT) primers. Differently barcoded cDNAs were pooled together, and second-strand cDNA was synthesised using the nick translation method. Next, libraries were tagmented with Tn5 transposase, and the 3'cDNA fragments were amplified with Illumina-compatible primers[49]. Paired-end sequencing was performed on the Illumina NextSeq 500. Alignment and read demultiplexing were performed using STAR (v2.7.8a)[82] and BRBseqTools (v 1.6)[86], respectively. Differentially expressed genes were identified using DESeq2 with a cutoff of FDR < 0.05 and FC > 1.5.

## Fractionation

Chromatin RNA extraction was performed using the described method[7]. 100 mg of siliques were ground in liquid nitrogen and resuspended in 20 ml of cold Honda Buffer (20 mM HEPES-KOH pH 7.4, 0.44 M sucrose, 1.25% Ficoll, 2.5% Dextran T40, 10 mM MgCl₂, 5 mM DTT, 0.5% Triton X−100, 10 mM β-mercaptoethanol, 1 mM PMSF,

1x Complete protease inhibitors (Roche), and 5 U/ml murine RNase inhibitor (NEB)). After rotation at 4 °C for 10 min and filtration through Miracloth, samples were centrifuged at 2000 g for 15 min at 4 °C. The nuclei pellet was washed twice with 5 ml of Honda buffer. The pellet was resuspended in 600 μl of Honda buffer and purified on a 40%-75% Percoll density gradient by centrifugation at 10,000 g for 30 min at 4 °C. Purified nuclei were collected from the Percoll interface, washed with Honda buffer and resuspended in 500 μl of cold glycerol buffer (20 mM Tris-HCl pH 8.0, 75 mM NaCl, 0.5 mM EDTA, 0.85 mM DTT, 50% glycerol, 1% Empigen, 10 mM β-mercaptoethanol, 0.125 mM PMSF, 1x Complete protease inhibitors (Roche), and 5 U/ml murine RNase inhibitor (NEB)). Samples were overlaid on urea lysis buffer (10 mM HEPES-KOH pH 7.4, 7.5 mM MgCl₂, 0.2 mM EDTA,300 mM NaCl, 1 M urea, 1% NP-40, 10 mM β-mercaptoethanol, 0.5 mM PMSF, 1x Complete protease inhibitors (Roche), and 5U/ml murine RNase inhibitor (NEB), vortexed for 2 s, and incubated on ice for 30 min, followed by centrifugation at 20,000 g for 2 min at 4 °C. The chromatin pellet was washed twice with 600 μl of urea lysis buffer for 30 min on a rotator at 4 °C. Chromatin pellet was used for RNA extraction and next for DNAse treatment as described above. The chromatin-attached RNA quality was tested by agarose gel electrophoresis and quantified with a Nanodrop 2000 spectrophotometer. cDNA synthesis was performed using a SuperScript III kit (Invitrogen).

**smFISH.** smFISH assay was performed as described previously[7] using a manually dissected radicle from dry embryos. The Stellaris probes used for smFISH target the full *shDOG1* sequence, including intron 1 and are labelled with Quasar670 fluorophore (Biosearch Technologies). The embryos were fixed, permeabilised and hybridised with the probes as described[7]. The smFISH signals were gathered using a widefield fluorescence microscope Olympus IX81 (Olympus) and a Hamamatsu Orca-R2 (C10600) CCD camera. The xCellence software (Olympus) was used for image acquisition. Cells were manually segmented using Napari[87] and the PartSeq plugin, which allows foci identification and classification based on subcellular location[88]. Dots number and intensity were analysed using R scripts described earlier[7].

## Statistics & reproducibility

All statistical analyses were performed using RStudio or Excel. Sample sizes are mentioned in figure legends and/or displayed in plots as individual data points. Sample sizes were not pre-specified. Information regarding the statistical test applied to each data is mentioned in the corresponding figure legend or source data. Significant differences were accepted at $p < 0.05$.

## Declaration of generative AI and AI-assisted technologies in the writing process

While preparing this work, the authors used Grammarly to improve readability and language. After using this tool, the authors reviewed and edited the content as needed and take full responsibility for the publication's content.

## Reporting summary

Further information on research design is available in the Nature Portfolio Reporting Summary linked to this article.

## Data availability

The 3'RNA-seq data generated for this study have been deposited at the Gene Expression Omnibus (GEO) under the accession code GSE274757. Source data are provided with this paper.

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

## Acknowledgements

We thank Krzysztof Kokoszka for plant cultivation and Suchismita Masanta for RNA isolation and genotyping. This work was funded by the Foundation for Polish Science (FNP TEAM POIR.04.04.00-00-3C97/16) and by the Polish National Science Centre (NCN SONATA BIS UMO-2018/30/E/NZ1/00354) awarded to S.S. S.P.S. acknowledges funding from the Polish National Science Centre (NCN OPUS 2024/53/B/NZ1/

03741), M.K. acknowledges funding from the Polish National Science Centre (NCN OPUS 2021/41/B/NZ3/02605 and OPUS 2024/53/B/NZ3/02252).

## Author contributions

S.P.S. and S.S. designed the experiments. S.P.S. carried out most of the experiments, M.M. and V.H.M. performed smFISH, and L.B. performed ChIDP-seq. M.K. conducted RNA-seq, K.R. performed 3'RACE-seq and obtained msh-1 x esp1 line, K.J.R. created Magic beads used in ChIRP-seq and ChIDP-seq, M.W. and R.Y. obtained CRISPR *DOG1* mutant, L.S. and A.J. developed *pU1-70K::U1-70K-GFP u1-70k* lines. S.P.S. and S.S. wrote the manuscript.

## Competing interests

K.J.R. and S.S. have equity interests in ElementZero Biolabs UG. S.S. also serve on the Scientific Advisory Board for ElementZero Biolabs UG. All other authors declare no competing interests.
