## [Transparent Peer Review file · Nature Communications]

Chromatin retained MUSHER lncRNA integrates ABA and DOG1 signalling pathways to enhance Arabidopsis seeds dormancy.

Corresponding Author: Dr Szymon Swiezewski

Version 0:

Reviewer comments:

Reviewer #1

(Remarks to the Author)

This is an interesting manuscript that presents considerable data on the role of a non-coding RNA, which they name MUSHER, that is involved in the regulation of seed dormancy in Arabidopsis. MUSHER is transcribed from a region at the 3' end of the DOG1 gene. Several mutant lines with disruptions in MUSHER showed reduced expression of DOG1 and reduced seed dormancy. Their extensive and well-documented experiments convincingly demonstrate that MUSHER affect the selection of a proximal cleavage polyadenylation site, resulting in an increase in the shorter of two DOG1 mRNAs. Additional assays show that MUSHER also interacts with the upstream sequences of the PIR1 gene, which is involved in ABA responsiveness, leading to increased transcription and ABA sensitivity. They also show that the interaction of MUSHER with chromatin is dependent on the U1 snRNP. Thus, they propose that MUSHER affects seed dormancy by increasing the expression of both DOG1 and PIR1 through two apparently distinct molecular pathways.

In my opinion, the research interpretations presented are well supported by evidence. Appropriate controls, confirmatory experiments and supporting data are presented. I do, however, have several serious questions about the presentation.

Major issues:

Both the Introduction section and the Discussion are highly redundant with the first paragraphs repeated almost word-for-word, in later sections. For example, the entire last half of the introduction is a repeat, nearly word for word, of the first half. Likewise, in the Discussion, the section following the subheading: DOG1-ABA pathways crosstalk is a largely a repeat of the first few paragraphs. Surely these sections require extensive editorial work.

Minor issues: (I note that neither the pages or the lines are numbered, making it difficult to identify the locations of the problematic sections.)

First page in Introduction, last paragraph. The statement "Even though dog1 mutants do not show changed ABA sensitivity," is a sentence fragment.

First page of Results. The statement: "T-DNA insertion lines aiming to disturb MUSHER transcriptional unit...." Should be disrupt the MUSHER transcriptional unit.

Third page of Results, second sentence. The statement: To test MUSHER localisation, we used developing seeds as seed dormancy is established during seed development, aka maturation, as a result of DOG1 and other genes expression. Abbreviation not defined: aka. Last phrase should be: as a result of the expression of DOG1 and other genes.

Second sentence in subsection titled: MUSHER lncRNA is required for DOG1 regulation by CPL1, a Pol II phosphatase. "Given MUSHER effect on DOG1 expression..." Should be Given the effect of MUSHER on DOG1 expression...

Second paragraph in subsection titled: MUSHER promotes DOG1 proximal polyadenylation by recruiting the CPSF complex. "...similarly as MUSHER knockouts... Should be: ...similarly to....

Third paragraph: ...by quantifying DOG1 transcription sites intensity...” Should be: by quantifying the intensity of DOG1 transcription sites..... Also, “...compared to WT (Figs. 3a,b). Should be (Supplementary Figs 3a,b).

Fourth paragraph, third sentence. “Contrary, some....” Should be: On the contrary, some... Or Conversely, some.

Subsection titled: MUSHER enhances secondary dormancy in response to high-temperature, second sentence. “ Based on msh mutant weak dormancy and MUSHER requirement for high DOG1 expression....” Should be: Based on the weak dormancy in musher mutants and the requirement for high DOG1 expression.

Third paragraph. “ Next, we set out to test....” Next, we tested....

Subsection titled: MUSHER enhances the PIR1 gene expression independently of its role in DOG1 regulation. First two sentences. What plants are being analyzed? According to the figure, msh1 vs WT. Should be stated in the text.

Subsection titled: MUSHER lncRNA activates PIR1 expression through PIR1-DOG1 chromatin interaction. First sentence and subsequent citations. “Fig.1” should be Fig. 1H. Also I could not find Supplementary Table 1.

Discussion, third paragraph. “...ABA role in secondary dormancy...” Should be: the role of ABA in secondary dormancy...

Discussion sixth paragraph. “...but using RNA-seq analysis indicates that...” Should be but RNA-seq analysis indicates that.... (delete using). “...often regulate gene expression at the chromatin.” ...at the level of chromatin. “...chromatin retention may be actively enforced.” Not sure what is meant by “enforced”. Actively determined?

Discussion, seventh paragraph. “...while RIR1 expression is nearly impossible in...” Should be: undetectable.

Again, text under the subheading: DOG1-ABA pathways crosstalk is largely duplicated from the beginning of the discussion.

Reviewer #2

(Remarks to the Author)

Reviewer #3

(Remarks to the Author)

In the study, the authors identify a chromatin-localized lncRNA - MUSHER, which activates two parallel genetic pathways and increases Arabidopsis seed dormancy. MUSHER could promote DOG1 expression by recruitment of the CPSF complex to enhance the proximal cleavage and polyadenylation of the DOG1 gene. It also increases seed ABA sensitivity by activating PIR1 gene transcription. They showed that, although these genes are located on different chromosomes, they are both bound by MUSHER. They further showed that MUSHER and other lncRNAs interact with U1 snRNP, and are required for their chromatin localization. The results suggest that U1's role in chromatin retention is not limited to a specific lncRNA, instead, it is likely evolutionarily conserved in plants. The study also showed that MUSHER promotes DOG1 proximal polyadenylation by locally affecting cleavage and polyadenylation complexes and Pol II occupancy at the proximal transcription termination site without affecting DOG1 transcription levels. The data together indeed provide new insight into the role of lncRNAs in plants.

However, I have several major concerns:

1. The manuscript mainly used two T-DNA insertion mutants for investigation. To make the data solid, more genetic evidences are needed, including the plants overexpressing the lncRNA, RNAi plants, and large fragment deletion plants. The crossover plants are also necessary.
2. The study was telling a story about a lncRNA regulating seed dormancy, however, germination tests only have a statistical chart of the germination rate (Fig1E, Fig3A, 3B, and Fig5E), there was not a figure or image showing the phenotype of seed germination or dormancy, this phenotype is necessary.
3. In Fig2B, 2E, 2G, Fig3B, 3D, Fig4C, 4D, Fig5E, Fig6B, 6E, 6F and Supplementary Fig2A, 2B, 2E, only one line of the mutants was used in the experiment and the T-DNA insertion mutant msh-1 was mainly used for observation. More lines are needed, including RNAi plants or overexpressing plants.
4. In Fig 5A, 5B ChIRP-seq showed MUSHER can bind to TSS of PIR1 and decreased H3K4me3 level. However, lncRNA does not have catalytic functions. How does it achieve this process? In addition, the authors showed that MUSHER could bind to different genes in different chromosomes although despite lacking sequence homology, what is the mode of action that the lncRNA binds to these genes?
5. ChIRP-seq revealed 23 putative DNA regions bound by MUSHER lncRNA, which is inconsistent with conventional NGS

results?

6. 3' RNA-seq analysis revealed 166 differentially expressed genes, except for DOG1, PIR1, are there any genes related to GA and ABA biosynthesis among the remaining differential genes? Are there any changes in the levels of ABA and GA in msh-1?

Some other concerns

1. The introduction section is redundant, with multiple repetitions regarding the citations on lncRNAs.
2. Fig 1D, the expression level of DOG1 in msh_dCas9-1 reduces to about 0.25, but in Fig1F the expression level of DOG1 almost not be detected, why?
3. Fig 2D, the figure had four lines but not including the legend, not consistent with the description in the text.
4. Fig 5C, 5D, the indicated position of PIRI probes in the figure is upstream of the gene. "One explanation is based on the chromatin-DNA precipitation assay (ChIDP) followed by sequencing, using PIRI DNA as a probe. This showed that PIRI DNA also has a very limited number of DNA targets" ChIDP-seq result should be shown.

Version 1:

Reviewer comments:

Reviewer #1

(Remarks to the Author)

This manuscript presents important new information about the role of non-coding RNAs in the regulation of gene expression in plants with specific emphasis on seed dormancy. The authors have addressed the reviewers' comments adequately and the data presentation and readability of the revised manuscript are much improved. I have noted a few grammatical errors yet to be corrected.

Line 38. DOG1 gene should be DOG1 transcript. It is the transcript that is cleaved and polyadenylated, not the gene.

Line 77. Cues, not clues.

Line 86. GWAs not defined. Should be: ...identified as a major QTL for seed dormancy and shown to have a significant association in genome wide studies.

Line 252, ...antagonistic effects... not effect.

Line 317.we performed ChIP-qPCR, not the ChIP-qPCR.

Line 598. sequesters not sequester.

Line 640. Importantly, MUSHER's association... or ...the association of MUSHER with chromatin...

Reviewer #2

(Remarks to the Author)

Reviewer #3

(Remarks to the Author)

The authors have addressed most of my concerns, however, there are some issues need to be addressed before the manuscript can be further consideration.

Specific comment as follows:

1. One of the major issues is the biological replicates of experiments, for example:

(1) Line 457, 3' RNA-seq analysis was performed between WT and msh-1 freshly harvested seeds. RNA-seq experiments require at least three biological replicates. The msh-1 line is a T-DNA insertion line, and the T-DNA insertion could affect local chromatin state changes, it is necessary to use multiple alleles for biological replicates to ensure the reliability of the results.

(2) Line 574, Similarly, the ChIRP experiment lacks biological replicates. The length column in the table does not correspond to the coordinate positions of the peaks.

(3) The alleles used in the experiment were not uniform. In Fig. 2d 2g 2h 2i 2j, some figures have the NoAB group, while others do not. Figure 2d includes msh-1 and msh-dCas9-1, but other figures lack corresponding experimental data.

MUSHER have five independent lines, but in Fig. 1g Fig. 2g 2h 2i 2j Fig. 4e 4f 4g 4h 4c Supplementary Fig. 3a 3b 3c 3d etc, only one line was used. These experimental data at least from two different types of mutants should be provided.

2. Another concern is the data issue:

(1) In Fig4a, the legend describes a fold change of 1.5, but in the method (line 985), it states "DESeq2 with a cutoff of FDR < 0.05 and FC > 1", not consistent. Although the authors provided the accession number for the 3' RNA-seq data, the data is inaccessible. Details of the differentially expressed genes should be provided.

(2) Fig. 3f, Fig. 5e, Fig. 6c, Supplementary Fig. 2a, Supplementary Fig. 6c - the error bar is very large such data cannot be directly used for a standard t-test, and relevant statistical principles should be followed when calculating the p-value. However, the authors claimed to have used a two-tailed, whereas unequal variances necessitate Welch's correction. For data with significant variations in the article, increasing the n to enhance the power of the test is important.

(3) Figure 1b, 1g; Figure 3c, 3e, 3f, 3h, 3g; Figure 6a, 6b, 6c; Supplementary Figure 2c, 2d; Supplementary Figure 3a, 3b, 3c. A large number of graphs lack p-value calculations. Please provide the p-value for each significance analysis.

(4) In Fig. 5e Supplementary Fig. 6 only one pir1 alleles was used. This is the key part of the article and more lines are needed. "We envisage that MUSHER may control H3K4me3 levels at PIR1 promoter through U1 snRNP interaction, which in turn facilitates Pol II recruitment that results in H3K4me3 increase, as strongly supported by existing literature data." Additional experimental data is required to support this conclusion, including the alterations in H3K4me3 levels in the absence of U1 or POLII and evidence of a dose-response effect between MUSHER and H3K4me3 levels.

3. The third is the control applied:

In Fig. 5a, LacZ-probes were used as a negative control; however, in ChIRP-seq, it appears that no negative control was employed. The standards of this experiment required not only the LacZ probe, but also the probes that should be divided into odd groups (Odd) and even groups (Even) to serve as controls, thereby reducing false positives. Additionally, the experiment should be conducted with three biological replicates.

4. Others

(1) The article describes the hybridization of msh-1 and pir1, but not conducted the hybridization of msh-1 and dog1, why? They are two parallel pathways and target genes. In addition, under the background of msh, overexpression of dog1/pir1 or their own promoters can be used to complement DOG1/ PIR1 to observe whether the level of dormancy is enhanced or restored.

(2) cpl1 mutant was used as a mimic of MUSHER over-expression line, MUSHER may control H3K4me3 levels at PIR1 promoter through U1snRNP interaction, which in turn facilitates Pol II recruitment that results in H3K4me3 increase. What I concerned is : CPL is very important, the CPL deficiency would reduce the overall transcription level, which conflicts with the study viewpoint that MUSHER does not affect the transcription rate of DOG1. Secondly, the H3K4me3 in the PIR1 promoter region requires Pol II to mediate, and the absence of CPL would directly lead to changes in H3K4me3 levels. We feel that using the direct overexpression of MUSHER is more appropriate.

(3) The RACE results should be provided. And the coding ability of MUSHER should be detected to exclude the possibility of encoding proteins or small peptides.

(4) In the point-by-point response, the authors mentioned that: "Fig. 6b, the ChIRP-seq data were validated not only in the msh-1 background but also using lacZ probes, which reinforces the robustness of these results." Fig. 6b is a RIP-qPCR experiment! In Fig. 2g, the position of the error bar is incorrect.

Version 2:

Reviewer comments:

Reviewer #3

(Remarks to the Author)

The authors have addressed my previous concerns and questions, thank you. Several minor issues need to clarify before it can be published.

Minor issues :

1. The Supplemental Table S5 showed the use of MACS3 to call peaks, but in the method, it was written as MACS2?

2. In the legend of Supplementary Fig. 1c and d, the letter 'c' was missing; the legend indicated 'MUSHER_2' and 'MUSHER_3', whereas the figure displayed 'MUSHER_22' and 'MUSHER_23'?
3. In Fig. 2g, the position of the error bar is incorrect.
4. Line 543-545 "Analysis of the ChIRP-seq data showed two replicates." However, the figure legend described that the data are the mean of three biological replicates?

Reviewer #1 (Remarks to the Author):

This is an interesting manuscript that presents considerable data on the role of a non-coding RNA, which they name MUSHER, that is involved in the regulation of seed dormancy in Arabidopsis. MUSHER is transcribed from a region at the 3' end of the DOG1 gene. Several mutant lines with disruptions in MUSHER showed reduced expression of DOG1 and reduced seed dormancy. Their extensive and well-documented experiments convincingly demonstrate that MUSHER affect the selection of a proximal cleavage polyadenylation site, resulting in an increase in the shorter of two DOG1 mRNAs. Additional assays show that MUSHER also interacts with the upstream sequences of the PIR1 gene, which is involved in ABA responsiveness, leading to increased transcription and ABA sensitivity. They also show that the interaction of MUSHER with chromatin is dependent on the U1 snRNP. Thus, they propose that MUSHER affects seed dormancy by increasing the expression of both DOG1 and PIR1 through two apparently distinct molecular pathways.

In my opinion, the research interpretations presented are well supported by evidence. Appropriate controls, confirmatory experiments and supporting data are presented. I do, however, have several serious questions about the presentation.

Thank you for your positive assessment of our manuscript. We appreciate your constructive criticism and carefully addressed your concerns to improve the clarity and presentation of our findings.

Major issues:

Both the Introduction section and the Discussion are highly redundant with the first paragraphs repeated almost word-for-word, in later sections. For example, the entire last half of the introduction is a repeat, nearly word for word, of the first half. Likewise, in the Discussion, the section following the subheading: DOG1-ABA pathways crosstalk is a largely a repeat of the first few paragraphs. Surely these sections require extensive editorial work.

Thank you for pointing this out. We apologize for the redundancy and appreciate the reviewer's careful reading. We removed the repeated fragments from the introduction and substantially revised the Discussion to better align with the presented data and improve the overall quality of the manuscript. The discussion is now divided into three main paragraphs:

1. **MUSHER-Mediated Regulation of *DOG1* and *PIR1* Expression.**
2. **MUSHER as a Link Between ABA Sensitivity and DOG1**
3. **U1 role in chromatin retention of lncRNA in Arabidopsis.**

Minor issues:

(I note that neither the pages or the lines are numbered, making it difficult to identify the locations of the problematic sections.)

We apologize for the oversight. We have now added line numbers to improve clarity.

First page in Introduction, last paragraph. The statement "Even though *dog1* mutants do not show changed ABA sensitivity," is a sentence fragment.

Corrected accordingly: Although *dog1* mutants do not exhibit altered ABA sensitivity²⁷, DOG1 protein has been shown to interact with and inhibit the enzymatic activity of PP2CA in vitro^{37,38}. (lines 110-112)

First page of Results. The statement: “T-DNA insertion lines aiming to disturb MUSHER transcriptional unit...” Should be disrupt the MUSHER transcriptional unit.

Corrected accordingly: To disrupt the *MUSHER* transcriptional unit, we used two T-DNA insertion lines, *msh-1* and *msh-2* (Fig. 1a), a Cas9 deletion mutant Δmsh , and two additional lines: *msh_dCas9-1* and *msh_dCas9-2* with catalytically inactive Cas9 (dCas9). (lines 139-142)

Third page of Results, second sentence. The statement: To test MUSHER localisation, we used developing seeds as seed dormancy is established during seed development, aka maturation, as a result of DOG1 and other genes expression. Abbreviation not defined: aka. Last phrase should be: as a result of the expression of DOG1 and other genes.

Corrected version: To test *MUSHER* localisation, we used developing seeds, as this is when *MUSHER* and *DOG1* are co-expressed at high levels, and *DOG1* exerts its function in seed dormancy establishment. (line 224)

Second sentence in subsection titled: MUSHER lncRNA is required for DOG1 regulation by CPL1, a Pol II phosphatase. “Given MUSHER effect on DOG1 expression...” Should be Given the effect of MUSHER on DOG1 expression...

Corrected: Given the effect of *MUSHER* on *DOG1* expression and dormancy, we speculated that CPL1 could act through *MUSHER* regulation. (line 243)

Second paragraph in subsection titled: MUSHER promotes DOG1 proximal polyadenylation by recruiting the CPSF complex. “...similarly as MUSHER knockouts... Should be: ...similarly to....

Corrected: We previously reported that mutations in pre-mRNA cleavage and polyadenylation factors result in reduced *shDOG1* but not *lgDOG1* expression¹, similar to *MUSHER* knockouts shown here. (line 324)

Third paragraph: ...by quantifying DOG1 transcription sites intensity...” Should be: by quantifying the intensity of DOG1 transcription sites..... Also, “...compared to WT (Figs. 3a,b). Should be (Supplementary Figs 3a,b).

We decided to rewrite this paragraph significantly to improve its clarity, including proposed changes: To test this hypothesis, we took an orthogonal approach and analysed the intensity of *DOG1* transcription using single-molecule RNA FISH (smFISH) in developing embryos². (line 348)

and

***DOG1* smFISH showed a consistent and significant reduction in the number of both nuclear and cytoplasmic foci during seed development in *msh-1* seeds when compared to WT (Supplementary Figs.3a,b). (line 355)**

Fourth paragraph, third sentence. “Contrary, some....” Should be: On the contrary, some... Or Conversely, some.

Corrected: Some lncRNAs may recruit specific histone modelling complexes, including PRC2 involved in H3K27me3 deposition. (line 368)

Subsection titled: MUSHER enhances secondary dormancy in response to high-temperature, second sentence. “Based on msh mutant weak dormancy and MUSHER requirement for high DOG1 expression...” Should be: Based on the weak dormancy in musher mutants and the requirement for high DOG1 expression.

Corrected as follows: Based on the seed’s requirement for *MUSHER* to express *DOG1* at high levels and fully develop primary dormancy (Fig. 1), we hypothesised that *MUSHER* may also be involved in secondary dormancy regulation. (line 380)

Third paragraph. “Next, we set out to test....” Next, we tested....

Corrected: Next, we tested if *MUSHER* accumulates in the cytoplasm or chromatin, which could provide insight into its mode of action. (line 223)

Subsection titled: MUSHER enhances the PIR1 gene expression independently of its role in DOG1 regulation. First two sentences. What plants are being analyzed? According to the figure, msh1 vs WT. Should be stated in the text.

Corrected as follows: 3’RNA-seq analysis revealed 166 differentially expressed genes, including *DOG1*, between WT and *msh-1* freshly harvested seeds. (line 466)

Subsection titled: MUSHER lncRNA activates PIR1 expression through PIR1-DOG1 chromatin interaction. First sentence and subsequent citations. “Fig.1” should be Fig. 1H. Also I could not find Supplementary Table 1.

Corrected: We showed that *MUSHER* is chromatin-bound (Fig. 1h). (line 532)

Discussion, third paragraph. “...ABA role in secondary dormancy...” Should be: the role of ABA in secondary dormancy...

Fragment was deleted.

Discussion sixth paragraph. “...but using RNA-seq analysis indicates that...” Should be but RNA-seq analysis indicates that.... (delete using).

Done. (line 754)

“...often regulate gene expression at the chromatin.” ...at the level of chromatin.

“...chromatin retention may be actively enforced.” Not sure what is meant by “enforced”. Actively determined?

Corrected as follows: Also, like their human counterparts, plant lncRNAs often regulate gene expression by chromatin binding³, indicating that their chromatin retention may be actively maintained. (line 755)

Discussion, seventh paragraph. "...while RIR1 expression is nearly impossible in..." Should be: undetectable.

Corrected: This view is supported by the fact that *DOG1* expression is reduced roughly by 50%, while *PIR1* expression is undetectable in multiple *msh* mutants tested. (line 705)

Again, text under the subheading: DOG1-ABA pathways crosstalk is largely duplicated from the beginning of the discussion.

The entire discussion section was substantially improved.

Thank you for your efforts to improve our manuscript we hope that you will find it ready for publishing in the current form.

Reviewer #2 (Remarks to the Author):

Reviewer #3 (Remarks to the Author):

In the study, the authors identify a chromatin-localized lncRNA - MUSHER, which activates two parallel genetic pathways and increases Arabidopsis seed dormancy. MUSHER could promote DOG1 expression by recruitment of the CPSF complex to enhance the proximal cleavage and polyadenylation of the DOG1 gene. It also increases seed ABA sensitivity by activating PIR1 gene transcription. They showed that, although these genes are located on different chromosomes, they are both bound by MUSHER. They further showed that MUSHER and other lncRNAs interact with U1 snRNP, and are required for their chromatin localization. The results suggest that U1's role in chromatin retention is not limited to a specific lncRNA, instead, it is likely evolutionarily conserved in plants. The study also showed that MUSHER promotes DOG1 proximal polyadenylation by locally affecting cleavage and polyadenylation complexes and Pol II occupancy at the proximal transcription termination site without affecting DOG1 transcription levels. The data together indeed provide new insight into the role of lncRNAs in plants.

Thank you for your thoughtful and positive evaluation of our study. We sincerely appreciate your recognition of our findings and their contribution to understanding lncRNA function in plants.

However, I have several major concerns:

comment:

1. The manuscript mainly used two T-DNA insertion mutants for investigation. To make the data solid, more genetic evidences are needed, including the plants overexpressing the lncRNA, RNAi plants, and large fragment deletion plants. The crossover plants are also necessary.

Thank you for your comment. To strengthen the robustness of our data, we generated a CRISPR/Cas9 mutant with a large deletion within the MUSHER transcriptional unit. This mutant exhibited a clear phenotype and reduced DOG1 expression, as shown in the manuscript (Fig. 1a and e). However, we also detected artificial transcription arising across the deletion (Supplementary Fig. 1c and d), which introduced potentially confounding effects. Consequently, we decided to exclude this line from further analysis. We'd like to kindly emphasise that key experiments were conducted using five independent lines disrupting MUSHER to ensure the robustness of our findings. Additionally, through various experiments, we demonstrated that inactivation of CPL1 (RNA polymerase II C-terminal domain phosphatase-like 1) leads to an overexpression of MUSHER, elevated expression of its targets, shDOG1 and PIR1, and increased dormancy in cpl1-7 compared to wild-type plants, suggesting that the cpl1 mutant can be used as a mimic of MUSHER over-expression line. We are fully aware of the pleiotropic phenotype of cpl1, however, we show clear and specific effect of several cpl1 alleles on MUSHER and shDOG1 in terms of both expression level and phenotype. In addition, genetic analysis shows that the CPL1 effect on DOG1 is MUSHER dependent. By analysing cpl1 we could assess the results of MUSHER's overexpression without the caveats created by introducing an ectopically overexpressed transgene.

Furthermore, we opt against using RNAi lines, whenever possible. In addition to post-transcriptional gene silencing, RNAi can induce chromatin modifications, potentially leading to broader epigenetic changes rather than simply downregulating the target

transcript. Given the proximity of *MUSHER* and *DOG1*-producing transcriptional units, this could obscure the interpretation of *MUSHER* depletion.

We do agree that additional crosses of the mutants presented in the original version will benefit our work. In the revised version of the manuscript, we have included *msh-1* crosses with *cpl1-7*, *esp-1* and *pir1*. All of them support the main conclusion that *MUSHER* controls seed dormancy through *shDOG1* and *PIR1*.

2. The study was telling a story about a lncRNA regulating seed dormancy, however, germination tests only have a statistical chart of the germination rate (Fig1E, Fig3A, 3B, and Fig5E), there was not a figure or image showing the phenotype of seed germination or dormancy, this phenotype is necessary.

We appreciate the reviewer's suggestion and have addressed this in the revised manuscript, we included representative photographs of germination tests, currently presented in Supplementary Fig. 1, 4, 6 and 7.

3. In Fig2B, 2E, 2G, Fig3B, 3D, Fig4C, 4D, Fig5E, Fig6B, 6E, 6F and Supplementary Fig2A, 2B, 2E, only one line of the mutants was used in the experiment and the T-DNA insertion mutant *msh-1* was mainly used for observation. More lines are needed, including RNAi plants or overexpressing plants.

Below we address each panel mentioned by the reviewer:

Fig. 2b shows *short/long DOG1* ratio based on RNA-seq; adding additional mutant lines would require additional RNA-seq analysis. We believe the conclusions drawn from RNA-seq data, are well-supported by data shown for multiple *MUSHER* alleles in Fig 2c.

Fig2e and g, in the revised version, we did not include additional alleles for this experiment, but we strongly believe that our ChIP experiment with @CPSF73 in two *MUSHER* alleles and *MUSHER* expression in *esp1* shown on 2f support genetic analysis shown in 2e.

Figs. 3b and 3d, include only one *cpl1* allele, *cpl1-7* mutant. We have now included additional data on multiple additional *cpl1* alleles in Supplementary Fig 2a,b.

Fig. 4c,d show *PIR1* expression in one *MUSHER* and one *DOG1* allele. In the revised version, we have included *PIR1* expression analysis in multiple *MUSHER* alleles (Supplementary Fig 5a) and an additional *DOG1* allele (Supplementary Fig 5b).

Fig. 5e, we conducted additional ABA sensitivity tests, demonstrating that *msh-1*, *msh-2*, *msh_dCas9-1*, and *msh_dCas9-2* mutants are less responsive to ABA (revised Fig. 5e and included as additional Supplementary Fig. 6). We also added data from another *dog1* mutant (*dog1-4*) and an *msh-1 x pir1* cross to further substantiate the findings.

Fig. 6b, the ChIRP-seq data were validated not only in the *msh-1* background but also using lacZ probes, which reinforces the robustness of these results.

Fig. 6e-f, while we have only analysed one allele for *UIC* and one for *UI-70K*, they, however, code for subunits of the same complex: U1 snRNP^{1,2} and, in our opinion, therefore strongly reinforce the conclusion.

Supplementary Fig. 2a,b in the revised version of the manuscript include multiple alleles of the mutants used.

Supplementary Fig. 2e shows the genetic interaction of *cpl1-7* with *msh-1*, and adding additional alleles would require a substantial amount of work.

We discussed the use of RNAi and *MUSHER* over-expressor lines above, but we would like to add that we included a CRISPR deletion line in the revised version of the manuscripts.

We hope that these additional alleles adequately address the Reviewers' concerns and strengthen the manuscript. We sincerely appreciate the Reviewers' suggestions.

4. In Fig 5A, 5B ChIRP-seq showed *MUSHER* can bind to TSS of *PIR1* and decreased H3K4me3 level. However, lncRNA does not have catalytic functions. How does it achieve this process?

Indeed, while *MUSHER* itself presumably lacks catalytic activity, our data suggest that it interacts with the U1 snRNP complex to remain associated with chromatin. Importantly, U1 snRNP has a well-documented role in splicing and transcriptional termination^{1,2}. In addition, U1 has also been shown to interact with Pol II¹⁻⁴.

We envisage that *MUSHER* may control H3K4me3 levels at *PIR1* promoter through U1 snRNP interaction, which in turn facilitates Pol II recruitment that results in H3K4me3 increase, as strongly supported by existing literature data⁵⁻⁸.

In addition, the authors showed that *MUSHER* could bind to different genes in different chromosomes although despite lacking sequence homology, what is the mode of action that the lncRNA binds to these genes?

ChIDP-seq analysis revealed that the proximal polyadenylation site of *shDOG1* interacts with the *PIR1* promoter region but not *MUSHER* coding DNA region. However, *MUSHER* transcription appears to be essential for the enhanced expression of both *shDOG1* and *PIR1*. While we currently do not fully understand how *MUSHER* specificity is achieved, we speculate in the revised discussion that the proximity of *DOG1* proximal polyA site and *PIR1* locus in the nucleus guide *MUSHER* to its targets.

5. ChIRIP-seq revealed 23 putative DNA regions bound by *MUSHER* lncRNA, which is inconsistent with conventional NGS results?

Thank you for raising this point. Our ChIRP-seq analysis identified 23 putative DNA regions bound by *MUSHER* lncRNA. The Reviewer is right that conventional ChIRIP-seq often results in hundreds of targets being identified. We believe that the low number of targets is due to the low background-to-noise ratio in our ChIRP-seq data, achieved through the use of MagIC beads as described in the methods section.

We have revised the manuscript to provide a clearer explanation of the technical strengths of our approach.

6. 3' RNA-seq analysis revealed 166 differentially expressed genes, except for *DOG1*, *PIR1*,

are there any genes related to GA and ABA biosynthesis among the remaining differential genes? Are there any changes in the levels of ABA and GA in *msh-1*?

Among the 166 differentially expressed genes in *msh-1*, we did not identify any genes directly involved in GA or ABA biosynthesis. However, there are indications that *MUSHER* inactivation may indirectly influence the GA/ABA balance:

1. bZIP16 targets among upregulated genes:

Using the Plant Transcription Factor Database⁹, we found that 12 out of 58 upregulated genes in *msh-1* are targeted by the bZIP16 transcription factor ($p < 1.054e-04$). bZIP16 is a positive regulator of seed germination, which is predominantly expressed in seeds. It promotes germination by regulating the expression of ABA- and GA-responsive genes¹⁰.

2. MYR2 targets among downregulated genes:

Additionally, 11 out of 107 downregulated genes in *msh-1* are targeted by the MYR2 transcription factor ($p < 2.395e-06$). MYR2 influences levels of gibberellins by regulating the expression of *GA20ox* and *GA3ox* genes. Notably, MYR1 and MYR2 regulate a complex feedback loop involving the simultaneous upregulation of GA biosynthesis and *DELLA* genes, as observed in *myr1 myr2* seedlings¹¹. This feedback loop complicates the interpretation of MYR2's precise role, making the relationship between *MUSHER* and MYR2 intriguing but beyond the scope of this manuscript.

3. bZIP1 and JAZ1:

We also observed upregulation of *bZIP1* and *JAZ1* in *msh-1* mutants. Both genes may be involved in ABA regulation¹²⁻¹⁴.

Thus, while no differentially expressed genes in *msh-1* are directly involved in GA or ABA biosynthesis, the transcriptional changes suggest indirect effects on GA/ABA dynamics. In our opinion, assessing hormone levels and their broader implications in *MUSHER*'s role in seed biology requires additional investigation, which falls outside the current study's focus.

Some other concerns

1. The introduction section is redundant, with multiple repetitions regarding the citations on lncRNAs.

Thank you for your comment. We have revised the manuscript to address this concern, minimizing redundancy in the Introduction and improving the overall clarity and quality of the text.

2. Fig 1D, the expression level of *DOG1* in *msh_dCas9-1* reduces to about 0.25, but in Fig1F the expression level of *DOG1* almost not be detected, why?

The data presented in Figs. 1d and 1f are from two independent experiments and reflect natural biological variation, including differences in *DOG1* induction in wild-type plants. To ensure accuracy and minimise artefacts related to growth conditions, we compared each sample to its respective side-by-side grown control.

3. Fig 2D, the figure had four lines but not including the legend, not consistent with the description in the text.

Thank you for raising this point. To maintain graphical clarity in Figs. 2d, f, h, i, and j, we opted to use a single legend positioned below the chart in Fig. 2j. This approach ensures consistency and avoids redundancy. We believe that in the improved version of the manuscript, the figure is fully aligned with the description in the text: “We observed a localised decrease in CPSF73 occupancy at the *DOG1* proximal but not distal termination site in *msh-1* and *msh_dCas9-1* mutants.”

4. Fig 5C, 5D, the indicated position of PIR1 probes in the figure is upstream of the gene. “One explanation is based on the chromatin-DNA precipitation assay (ChIDP) followed by sequencing, using PIR1 DNA as a probe. This showed that PIR1 DNA also has a very limited number of DNA targets” ChIDP-seq result should be shown.

Thank you for this comment. We acknowledge the importance of providing additional data for the ChIDP-seq experiment. In the revised manuscript, we have included the *PIR1* ChIDP target list in the Supplementary Table 2.

References

1. So, B. R. *et al.* A Complex of U1 snRNP with Cleavage and Polyadenylation Factors Controls Telescripting, Regulating mRNA Transcription in Human Cells. *Molecular Cell* **76**, 590-599.e4 (2019).
2. Mangilet, A. F. *et al.* The Arabidopsis U1 snRNP regulates mRNA 3'-end processing. *Nat. Plants* **10**, 1514–1531 (2024).
3. Zhang, S. *et al.* Structure of a transcribing RNA polymerase II–U1 snRNP complex. *Science* **371**, 305–309 (2021).
4. Chi, B. *et al.* Interactome analyses revealed that the U1 snRNP machinery overlaps extensively with the RNAP II machinery and contains multiple ALS/SMA-causative proteins. *Sci Rep* **8**, 8755 (2018).
5. Nielsen, M. *et al.* Transcription-driven chromatin repression of Intragenic transcription start sites. *PLoS Genet* **15**, e1007969 (2019).
6. Shu, J., Ding, N., Liu, J., Cui, Y. & Chen, C. Transcription elongator SPT6L regulates the occupancies of the SWI2/SNF2 chromatin remodelers SYD/BRM and nucleosomes at transcription start sites in Arabidopsis. *Nucleic Acids Research* **50**, 12754–12767 (2022).
7. Antosz, W. *et al.* The Composition of the Arabidopsis RNA Polymerase II Transcript Elongation Complex Reveals the Interplay between Elongation and mRNA Processing Factors. *Plant Cell* **29**, 854–870 (2017).
8. Pekowska, A. *et al.* H3K4 tri-methylation provides an epigenetic signature of active enhancers: Epigenetic signature of active enhancers. *The EMBO Journal* **30**, 4198–4210 (2011).
9. Tian, F., Yang, D.-C., Meng, Y.-Q., Jin, J. & Gao, G. PlantRegMap: charting functional regulatory maps in plants. *Nucleic Acids Research* gkz1020 (2019) doi:10.1093/nar/gkz1020.
10. Hsieh, W.-P., Hsieh, H.-L. & Wu, S.-H. Arabidopsis bZIP16 Transcription Factor Integrates Light and Hormone Signaling Pathways to Regulate Early Seedling Development. *Plant Cell* **24**, 3997–4011 (2012).
11. Zhao, C., Hanada, A., Yamaguchi, S., Kamiya, Y. & Beers, E. P. The Arabidopsis Myb genes *MYR1* and *MYR2* are redundant negative regulators of flowering time under decreased light intensity. *The Plant Journal* **66**, 502–515 (2011).
12. Ju, L. *et al.* JAZ proteins modulate seed germination through interaction with ABI 5 in bread wheat and Arabidopsis. *New Phytologist* **223**, 246–260 (2019).
13. Sun, X.-L. *et al.* Arabidopsis bZIP1 Transcription Factor Binding to ABRE cis-Element Regulates Abscisic Acid Signal Transduction. *Acta Agronomica Sinica* **37**, 612–619 (2011).
14. Fu, J. *et al.* OsJAZ1 Attenuates Drought Resistance by Regulating JA and ABA Signaling in Rice. *Front. Plant Sci.* **8**, 2108 (2017).

RESPONSE TO REVIEWER COMMENTS

Reviewer #1 (Remarks to the Author):

This manuscript presents important new information about the role of non-coding RNAs in the regulation of gene expression in plants with specific emphasis on seed dormancy. The authors have addressed the reviewers' comments adequately and the data presentation and readability of the revised manuscript are much improved. I have noted a few grammatical errors yet to be corrected.

Line 38. DOG1 gene should be DOG1 transcript. It is the transcript that is cleaved and polyadenylated, not the gene.

Line 77. Cues, not clues.

Line 86. GWAs not defined. Should be: ...identified as a major QTL for seed dormancy and shown to have a significant association in genome wide studies.

Line 252, ...antagonistic effects... not effect.

Line 317.we performed ChIP-qPCR, not the ChIP-qPCR.

Line 598. sequesters not sequester.

Line 640. Importantly, MUSHER's association... or ...the association of MUSHER with chromatin...

Response: Thank you for your positive opinion about our work. We corrected the grammatical errors and appreciate your positive comments on our revised manuscript.

Reviewer #2 (Remarks to the Author):

Reviewer #3 (Remarks to the Author):

The authors have addressed most of my concerns, however, there are some issues need to be addressed before the manuscript can be further consideration.

Response: Thank you for this comment. Indeed, we feel that, thanks to your points raised in the last revision, we have provided the requested additional CRISPR mutant alleles, which greatly enhance the coherence and quality of the manuscript.

Specific comment as follows:

1. One of the major issues is the biological replicates of experiments, for example:

(1) Line 457, 3' RNA-seq analysis was performed between WT and *msh-1* freshly harvested seeds. RNA-seq experiments require at least three biological replicates. The *msh-1* line is a T-DNA insertion line, and the T-DNA insertion could affect local chromatin state changes, it is necessary to use multiple alleles for biological replicates to ensure the reliability of the results.

Response: RNA-seq experiments were conducted with four biological replicates for both WT and *msh-1* freshly harvested seeds. This has been described in the previous version in the method section and the GEO data files. In the revised version, we included a PCA analysis of individual RNA-seq repeats (Supplementary Fig. 2f). As PCA reflects the expression of a few hundred of the most variable genes, relatively weak grouping of biological replicates reflects limited but specific differences associated with the *MUSHER* role in dormancy regulation. The reviewer is right that our setup does not allow us to exclude the indirect effect of the *msh-1* T-DNA allele. However, we would like to stress that the effect of *MUSHER* on the two main targets analysed in the manuscript, *DOG1* and *PIRI*, was validated using multiple *MUSHER* alleles, as shown in Fig. 1e, 2c, 3a, 4c-d, S1, S2a-b, S4a, S5a-b, and S6e-g.

Moreover, it is a common practice to conduct RNA-seq analysis in plant manuscripts using only one allele, including multiple prominent papers published within the last two years in Nature Communications:

1 10.1038/s41467-024-46082-5 *hen2* RNA-seq

2 10.1038/s41467-025-59095-5 *arf6 arf8* and *spy-3* RNA-seq

3 10.1038/s41467-025-57806-6 *cdk8* RNA-seq

(2) Line 574, Similarly, the ChIRP experiment lacks biological replicates. The length column in the table does not correspond to the coordinate positions of the peaks.

Response: We apologize for the oversight regarding the ChIRP experiment deposition. The ChIRP experiment was done in 2 replicates, and a PCA that allows comparison of replicate reproducibility is included in the revised Figure S6b. In addition to allowing easier assessment of the reproducibility, we have included single-replicate data tracks in the revised Figure S6a. Below, we also show the positions of peaks identified in ChIRP-seq experiments, along with read density profiles (bigWig tracks) of ChIRP-seq in WT and *msh-1*." Regarding the mismatch in the length column and coordinate positions, we regret the technical error caused by an Excel upload issue, which was not double-checked. In the revised submission, we have corrected the length column in the supplementary table S5 presenting ChIRP-seq results.

We would like to stress that ChIRP experiments are often done with two biological replicates, including:

1. 10.15252/embj.2022110921 a 2023 FAIL lncRNA ChIRP analysis
2. 10.1093/nar/gkaf093 a 2025 DUBR lncRNA ChIRP analysis
3. 10.1016/j.molcel.2019.12.015 a 2020 APOLO lncRNA ChIRP analysis

(3) The alleles used in the experiment were not uniform. In Fig. 2d 2g 2h 2i 2j, some figures have the NoAB group, while others do not.

Response: We appreciate your identifying our oversight. We updated Fig. 2h and 4f with NoAB controls. Since Figures 2h–j and 4f–h belong to the same experimental set, the NoAB control is presented only in Figures 2h and 4f, which collectively evaluate downstream H3K27me3 and H3K4me3 modifications. We have updated the corresponding Figure legends to clarify these points.

Figure 2d includes *msh-1* and *msh-dCas9-1*, but other figures lack corresponding experimental data.

MUSHER have five independent lines, but in Fig. 1g Fig. 2g 2h 2i 2j Fig. 4e 4f 4g 4h 4c Supplementary Fig. 3a 3b 3c 3d etc, only one line was used. These experimental data at least from two different types of mutants should be provided.

Response: We'd like to emphasize that key experiments were conducted using multiple independent lines disrupting *MUSHER* to ensure the robustness of our findings.

Fig. 1g

We incorporated an additional *msh-1* allele as Supplementary Fig. 1f.

Fig. 2g,h,i,j and Fig. 4e 4f 4g 4h

We used one *msh* allele for ChIP-qPCR. We appreciate the comments that multiple alleles would strengthen the conclusion, but it is a commonly accepted practice to do ChIP-qPCR in one allele of the mutant, including the following manuscripts from the last couple of years:

1. 10.1038/s41467-025-59268-2, pMYB5:MYB5-GFP/myb5-2 ChIP one line (Fig. 4f)
2. 10.1038/s41467-022-32897-7, RNA ChIP Pol II in one allele (Fig. 3a) and H3K27me3 (Fig. 3b)
3. 10.1038/s41467-023-37805-1, H3K9Ac ChIP in one allele (Fig. 6c)
4. 10.1038/s41467-025-59095-5 P_{UBQ10}:FLAG-ARF6/ARF8 single lines (Fig. 5c,d)
5. 10.1038/s41477-025-01962-6, H3K36me3 ChIP in one allele (Fig. S1d,2e,f)
6. 10.1038/s41467-025-57394-5pUBQ10:GFP-CHR11 single line (Fig. 4a,b)
7. 10.1038/s41467-025-57806-6, H3K9me2 ChIP in one allele (Fig. 8g)
8. 1038/s41477-022-01125-x H3 RIP in one allele (Fig. 2f)

Fig. 4c Describes the PIR1 expression in one *dog1* and one *msh* allele. In the previous revision, we have included one additional *dog1* and 4 additional *msh* alleles, see SFig. 5a-b

Supplementary Fig. 3a 3b 3c 3d

Data describe smFISH data. smFISH is still not a widely used method, but all known to us instances of smFISH use in plants involved only one allele of the mutant:

1. 10.15252/embj.2022112443 – one *puppies* knockout and one *PUPPIES* ox line.
2. 10.1038/ncomms13031 – one allele of *FLC-TEX* mutant.

2. Another concern is the data issue:

(1) In Fig4a, the legend describes a fold change of 1.5, but in the method (line 985), it states "DESeq2 with a cutoff of FDR < 0.05 and FC > 1", not consistent.

Response: Thank you for spotting our error. We used a fold change (FC) cut-off of >1.5, not >1. This has now been corrected in the revised method section.

Although the authors provided the accession number for the 3' RNA-seq data, the data is inaccessible.

Response: We are unsure where the problem is, but the data was deposited according to Nature Communications policy at the time of original submission and is accessible using the reviewer token provided in the original submission (GEO accession GSE274757; token: afmhssqgpjutryj). We can access it using this token.

Details of the differentially expressed genes should be provided.

Response: Thank you for spotting this omission. The differentially expressed genes are now provided in the Supplementary data table (Supplementary Table 3).

(2) Fig. 3f, Fig. 5e, Fig. 6c, Supplementary Fig. 2a, Supplementary Fig. 6c - the error bar is very large such data cannot be directly used for a standard t-test, and relevant statistical principles should be followed when calculating the p-value. However, the authors claimed to have used a two-tailed test, whereas unequal variances necessitate Welch's correction. For data with significant variations in the article, increasing the n to enhance the power of the test is important.

Response: We acknowledge the concern regarding large and unequal error bars in data sets in Fig. 3f, Fig. 5e, Fig. 6c, Supplementary Fig. 2a, and Supplementary Fig. 6c, which may indicate unequal variance.

We have re-evaluated the data using Welch's t-test where applicable, ensuring compliance with relevant statistical principles. For Supplementary Fig. 2a, we conducted Levene's test to assess variance homogeneity across WT, *cpl1-7*, *cpl1-8*, and *cpl1-9*, yielding $W = 0.9046$, $p = 0.4640$. As $p > 0.05$, the assumption of equal variances is satisfied, justifying the use of a standard t-test for this figure. The results of all the tests are provided in the Source Data file and described in the figure legend.

(3) Figure 1b, 1g; Figure 3c, 3e, 3f, 3h, 3g; Figure 6a, 6b, 6c; Supplementary Figure 2c, 2d; Supplementary Figure 3a, 3b, 3c. A large number of graphs lack p-value calculations. Please provide the p-value for each significance analysis.

Response: We used asterisks to indicate statistical significance. The source data provides full statistical details, including precise p-values for all relevant comparisons mentioned in the manuscript.

(4) In Fig. 5e Supplementary Fig. 6 only one *pir1* alleles was used. This is the key part of the article and more lines are needed.

Response: We thank the reviewer for the comment. While we agree it would be advantageous to use more *PIRI* alleles, we would, however, like to clarify that the enhanced sensitivity of *PIRI* mutants to ABA was originally reported using two alleles (10.1111/tbj.14507), and in this manuscript, we used one *PIRI* allele to reconfirm the authors' original conclusion.

"We envisage that MUSHER may control H3K4me3 levels at *PIR1* promoter through U1 snRNP interaction, which in turn facilitates Pol II recruitment that results in H3K4me3 increase, as strongly supported by existing literature data." Additional experimental data is required to support this conclusion, including the alterations in H3K4me3 levels in the absence of U1 or POLII and evidence of a dose-response effect between MUSHER and H3K4me3 levels.

Response: We thank the reviewer for the suggestion. While we acknowledge that additional experiments proposed by the Reviewer would further support the proposed model, we strongly believe that at this stage our data are mature for publishing and validation of the U1-H3K4me3 modulation by *MUSHER* is beyond the current manuscript.

We would definitely like to explore this direction in the future, especially given that this model is consistent with published molecular frameworks of U1 activity in animals, linking U1 SnRNP and Pol II recruitment to H3K4me3 level regulation.

3. The third is the control applied:

In Fig. 5a, LacZ-probes were used as a negative control; however, in ChIRP-seq, it appears that no negative control was employed. The standards of this experiment required not only the LacZ probe, but also the probes that should be divided into odd groups (Odd) and even groups (Even) to serve as controls, thereby reducing false positives. Additionally, the experiment should be conducted with three biological replicates.

Response: ChIRP-seq and ChIDP-seq were conducted with two biological replicates and showed high reproducibility, as demonstrated by the PCA plots included in the revised manuscript (Supplementary Fig. 6a-b). We have also included single-repeat data tracks for evaluation of the ChIRP and ChIDP-seq reproducibility (Supplementary Fig. 6a) and ChIDP-qPCR experiments (Supplementary Fig. 6c-d).

Many recently published studies rely on two replicates of genome-wide ChIRP, e.g.:

1. 10.1038/s41467-024-52783-8
2. 10.1093/nar/gkaf093

including plant manuscripts:

1. 10.15252/embj.2022110921
2. 10.1016/j.molcel.2019.12.015
3. 10.1093/plcell/koac334

While some ChIRP-seq protocols recommend dividing probes into Odd and Even subsets to further validate specificity and reduce potential false positives, this approach is not universally used. In our study, we used LacZ probes as a negative control, a widely accepted method for assessing background signal and nonspecific interactions. Additionally, we performed ChIRP-seq in the *msh-1* mutant background, providing an extra layer of negative control, reducing the possibility of false-positive signals related to using *MUSHER* probes.

Importantly, Odd/Even probe strategies are not strictly used in all ChIRP-seq experiments, as also supported by the following examples:

- 10.1038/s41467-023-40883-w
- 10.1093/plcell/koac334

For the ChIDP-seq, our initial qPCR results showed extreme specificity for the probes and method used, with nearly no detectable amplification in the empty beads control, as shown below. As a result, we decided not to use the empty beads control for the sequencing. We have now included the ChIDP-qPCR results, performed using six biological repeats, in the SFig. 6c-d as additional validation of the ChIDP-seq method.

4. Others

(1) The article describes the hybridization of *msh-1* and *pir1*, but not conducted the hybridization of *msh-1* and *dog1*, why? They are two parallel pathways and target genes. In addition, under the background of *msh*, overexpression of *dog1/pir1* or their own promoters can be used to complement *DOG1/PIR1* to observe whether the level of dormancy is enhanced or restored.

Response: We did not perform a *msh-1 dog1* double mutant analysis because *MUSHER*'s effect on *DOG1* regulation was already strongly supported by molecular data, including chromatin binding and polyadenylation control. In the revised manuscript, we further demonstrate that *MUSHER*-complemented lines (*msh-1* pMUSHER_1 and *msh-1* pMUSHER_2) show significantly elevated expression of *shDOG1* and *PIR1*, along with enhanced dormancy compared to *msh-1*, closely resembling wild-type plants (Supplementary Figs. 1g-i and 5c). We believe this provides strong functional evidence that *MUSHER* regulates both *DOG1* and *PIR1*. While additional genetic experiments, such as *DOG1* or *PIR1* overexpression in the *msh-1* background, could offer complementary insights, we consider the current data sufficient to support the conclusions of this study.

(2) *cpl1* mutant was used as a mimic of *MUSHER* over-expression line, *MUSHER* may control H3K4me3 levels at *PIR1* promoter through U1snRNP interaction, which in turn facilitates Pol II recruitment that results in H3K4me3 increase. What I concerned is : CPL is very important, the CPL deficiency would reduce the overall transcription level, which conflicts with the study viewpoint that *MUSHER* does not affect the transcription rate of *DOG1*. Secondly, the H3K4me3 in the *PIR1* promoter region requires Pol II to mediate, and the absence of CPL would directly lead to changes in H3K4me3 levels. We feel that using the direct overexpression of *MUSHER* is more appropriate.

Response: We thank the reviewer for their insightful comments and partially agree with their concerns. To further validate our conclusions, we have included *msh-1* *MUSHER*-complemented lines (*msh-1* pMUSHER_1 and *msh-1* pMUSHER_2) in our revised manuscript (Supplementary Fig. 1g-i and 5c). Additionally, we have revised the title of Supplementary Fig. 2 from "*MUSHER* overexpression activates *DOG1*" to "*cpl1-7* mimics *MUSHER* overexpression leading to upregulation of *DOG1*" to better reflect our findings. However, our data demonstrate that *MUSHER* lncRNA is a direct target of CPL1, a Pol II phosphatase, and is required for CPL1-mediated suppression of *DOG1* expression (Supplementary Fig. 2a-d). Specifically, *cpl1* mutants show elevated *MUSHER* levels, mimicking *MUSHER* overexpression, and *MUSHER* deletion suppresses *DOG1* upregulation in *cpl1* plants (Supplementary Fig. 2c). While CPL1 deficiency may broadly affect transcription, our results indicate that *MUSHER*, not CPL1, directly influences *DOG1* polyadenylation site selection, particularly the *shDOG1* isoform, without altering *DOG1* transcription rates (Supplementary Fig. 2e). Additionally, the reviewer's concern about H3K4me3 levels at the *PIR1* promoter overlooks our finding that *MUSHER*'s role in *DOG1* regulation is independent of CPL1's effect on Pol II-mediated H3K4me3. Additionally, there is no strong link between *cpl1* inactivation and H3K4me3 levels as suggested by the reviewer as H3K4me3 levels are unaffected at one of *MUSHER* targets - *DOG1* (Fig. 2i). We believe that the *cpl1* mutant remains a valid model for studying *MUSHER*'s effects, strongly supported by direct *MUSHER* complementation experiments, confirming the robustness of our findings.

(3) The RACE results should be provided.

Response: We provided detailed RACE results in Supplementary Table 1.

The coding ability of *MUSHER* should be detected to exclude the possibility of encoding proteins or small peptides.

Response: We performed an analysis using CPC (Coding Potential Calculator), which clearly shows that *MUSHER* doesn't have a coding potential similar to FLAIL, SVALK, ELENA, or APOLO. (Supplementary Table 2).

(4) In the point-by-point response, the authors mentioned that: "Fig. 6b, the ChIRP-seq data were validated not only in the *msh-1* background but also using lacZ probes, which reinforces the robustness of these results." Fig. 6b is a RIP-qPCR experiment! In Fig. 2g, the position of the error bar is incorrect.

Response: Thank you for pointing this out. That was our mistake—the correct reference should be to Fig. 6a and 6c. The error bars have now been corrected.

REVIEWERS' COMMENTS

Reviewer #3 (Remarks to the Author):

The authors have addressed my previous concerns and questions, thank you. Several minor issues need to clarify before it can be published.

Thank you. We appreciate your thorough review and valuable insights.

Minor issues :

1. The Supplemental Table S5 showed the use of MACS3 to call peaks, but in the method, it was written as MACS2?

Corrected.

2. In the legend of Supplementary Fig. 1c and d, the letter 'c' was missing; the legend indicated 'MUSHER_2' and 'MUSHER_3', whereas the figure displayed 'MUSHER_22' and 'MUSHER_23'?

Corrected.

3. In Fig. 2g, the position of the error bar is incorrect.

We have thoroughly reanalysed the data for Fig. 2g and identified a discrepancy in the positioning of the second error bar from the left. This has been corrected. To confirm, we have included a screenshot of an overlaid chart generated in Excel, which aligns with the updated Fig. 2g. Please let us know if this fully addresses the issue or if further clarification is needed.

4. Line 543-545 "Analysis of the ChIRP-seq data showed two replicates." However, the figure legend described that the data are the mean of three biological replicates?

Corrected.